# A 2.2 Å cryoEM structure of a quinol-dependent NO Reductase shows close similarity to respiratory oxidases

Alex J. Flynn[1,2,4], Svetlana V. Antonyuk [3,4], Robert R. Eady[3], Stephen P. Muench [1,2] ✉ & S. Samar Hasnain [3] ✉

Quinol-dependent nitric oxide reductases (qNORs) are considered members of the respiratory heme-copper oxidase superfamily, are unique to bacteria, and are commonly found in pathogenic bacteria where they play a role in combating the host immune response. qNORs are also essential enzymes in the denitrification pathway, catalysing the reduction of nitric oxide to nitrous oxide. Here, we determine a 2.2 Å cryoEM structure of qNOR from *Alcaligenes xylosoxidans*, an opportunistic pathogen and a denitrifying bacterium of importance in the nitrogen cycle. This high-resolution structure provides insight into electron, substrate, and proton pathways, and provides evidence that the quinol binding site not only contains the conserved His and Asp residues but also possesses a critical Arg (Arg720) observed in cytochrome *bo₃*, a respiratory quinol oxidase.

Quinol-dependent Nitric Oxide Reductases (qNORs) belong to a broader family of nitric oxide reductases (NORs). These integral membrane enzymes are an essential component of bacterial nitrate respiration, so-called denitrification which functions in oxygen-limited environments. In this process, nitrate is reduced to N₂ via a series of intermediates catalyzed by distinct oxidoreductases $NO_3^- \rightarrow NO_2^- \rightarrow NO \rightarrow N_2O \rightarrow N_2$. Located in the cytoplasmic membrane, NORs reduce the NO produced by nitrite reductase and catalyze N–N bond formation in the reaction $2NO + 2H^+ + 2e^- \rightarrow N_2O + H_2O$ with the formation and release of nitrous oxide (N₂O). This ozone-depleting and greenhouse gas is some 300-fold more potent than CO₂, and as such, has a strong negative environmental impact.

Three types of respiratory bacterial NOR are recognized that differ in the nature of their physiological electron donor, using either soluble cytochromes, pseudoazurin, or quinol. These are cytochrome *c*-dependent NOR (cNOR), quinol-dependent NOR (qNOR), and a novel NOR from *Bacillus azotoformans* that has a di copper center (Cu_A) which accepts electrons from cytochrome *c₅₅₁* (Cu_ANOR). Both qNOR and Cu_ANOR are electrogenic, conserving energy from NO reduction by creating a proton electrochemical gradient across the membrane[1,2]. In contrast, cNOR is not electrogenic and the protons required for the reduction of NO are taken up from the periplasm[3]. Whilst cNOR is predominantly found in denitrifying organisms, qNORs are also found in several pathogenic bacteria, where the enzyme detoxifies NO produced by the host's immune response[4,5].

The x-ray structures for representatives of both cNOR and qNOR have been determined[1,6,7], and in addition, cryoEM structures of two qNORs from the denitrifying pathogens *Neisseria meningitidis* and *Alcaligenes xylosoxidans* have been determined at 3.1 and 3.9 Å resolution, respectively[8,9]. Both cryoEM structures revealed dimeric assemblies in contrast to the monomeric assembly seen in the x-ray crystal structure of *Nm*qNOR, a difference attributed to the binding of zinc (used in the crystallization) near the predicted entry to the water channel and the dimer-stabilizing transmembrane TMII region[9]. Zinc was also shown to inhibit NOR activity of both *Ax* and *Nm* qNORs, consistent with the dimer being the active species. A non-Zn crystallization of *Ax*qNOR has led to a dimeric crystallographic structure but only to 6.5 Å[10,11]. Mutational studies of *Ax*qNOR (Glu494Ala) led to

[1]School of Biomedical Sciences, Faculty of Biological Sciences, University of Leeds, Leeds LS2 9JT, UK. [2]Astbury Centre for Structural and Molecular Biology, University of Leeds, Leeds LS2 9JT, UK. [3]Molecular Biophysics Group, Department of Biochemistry and Systems Biology, Institute of Systems, Molecular and Integrative Biology, Faculty of Health and Life Sciences, University of Liverpool, Liverpool L69 7ZB, England. [4]These authors contributed equally: Alex J. Flynn, Svetlana V. Antonyuk. ✉e-mail: S.P.Muench@leeds.ac.uk; s.s.hasnain@liverpool.ac.uk

similar destabilization of the dimeric assembly, resulting in a monomeric cryoEM structure with the mutant enzyme exhibiting a loss of enzymatic activity, demonstrating that activity is disrupted through the loss of dimeric assembly of qNORs[8,9].

We note that the abbreviation, cryoEM used throughout the manuscript, has evolved to indicate that **e**lectron **m**icroscopy (EM) is done with a sample maintained at **cryo**genic temperatures and distinguish it from the transmission electron microscopy without the cryogenic stage for the samples that are extensively used in material science research. We suggest that cryoEM should be standardized to stand for "single-particle **e**lectron **m**icroscopy with **cryo**genic sample stage (cryoEM)" as neither the microscope or electrons are at cryogenic temperatures, and as such, neither terms "cryo-electron microscopy" or "electron cryomicroscopy" are accurate.

Genomic sequence analysis has revealed NORs to be members of the respiratory heme-copper oxidase (HCO) superfamily that also has proton-pumping quinol and cytochrome $c$ oxidases members[12–14]. These families have a similar catalytic binuclear active site containing a high-spin heme ($b_3/a_3/o_3$) and either $Cu_B$ in the case of the $O_2$ reductases, or $Fe_B$ in the case of NORs. $Cu_B$ is coordinated by three conserved His residues, with the side chain of one covalently linked to the aromatic ring of a second-sphere Tyr residue implicated in electron and proton donation during catalysis. This link is missing in NORs and subfamilies show variable coordination of $Fe_B$[15]. The structurally characterized cNOR and qNORs have a $(His)_3$ or $(His)_3$-(Glu) primary ligation and a second-sphere Glu residue with a suggested proton-relay role.

cNORs consist of a complex of NorB and NorC, with the latter subunit containing heme $c$ that receives an electron from cytochrome and acts as the electron donor to the binuclear μ-oxo bridged heme $b_3$-$Fe_B$ catalytic center in NorB. In contrast, qNORs are single subunit enzymes (NorZ) which use membrane-bound quinol as an electron source. The C-terminus is highly homologous to NorB and the hydrophilic domain shows substantial structural homology for four of the five α-helices with the NorC subunit in cNOR, despite the loss of heme c and its binding motif (Cys-X-X-Cys-His). These findings have prompted the suggestion of gene-fusion in qNORs and the absence of a redox center (heme $c$) in this fused domain has provided further support to the idea that qNORs are related to the family of respiratory quinol oxidases.

Due to the difficulty in solubilizing and crystallizing membrane proteins, their structural determination remains challenging, despite the emergence of advanced synchrotron radiation sources[16], micro-focus beamlines[17,18], and improvement in purification and crystallization methods, including the lipidic cubic phase method[19]. Although cryoEM has accelerated membrane protein structure determination, for a long time, it was difficult to obtain resolutions high enough to visualize metalloenzyme chemistry in their redox centers (~2 Å).

Since publishing our previous $Ax$qNOR structure at 3.9 Å resolution in 2019, there have been advances in hardware for cryoEM, including detectors like the Falcon 4 with improved detective quantum efficiency (DQE), better signal-to-noise ratio (SNR) and faster collection rates. Furthermore, new energy filters like the Selectris are better at removing inelastically scattered electrons that contribute to noise, leading to a further improvement in SNR. Recently, a cryoEM setup containing the Falcon 4 and Selectris energy filter resulted in the highest resolution structure of the cryoEM test specimen apoferritin at 1.22 Å and the transmembrane β3 $GABA_A$ receptor at 1.7 Å[20]. These new hardware developments represent potentially useful tools for studying metalloenzymes and visualizing the detailed structural arrangements of the metal active sites at high resolution. This, combined with our demonstration by cryoEM that qNORs have a dimeric structure in solution[8,9] in contrast to the zinc-induced monomeric crystallographic structure[1,7], places cryoEM to be a promising approach to probe the structure–function relationship of metalloenzymes at high resolution.

Here, we report a high resolution, 2.2 Å, structure of $Ax$qNOR. This has provided a detailed view of a dimeric NOR molecule, which is considered to be the active form of qNORs. This resolution permits a clear definition of the binuclear active site where $Fe_B$ and heme $b_3$ that are linked by a μ-oxo bridge which in turn is hydrogen bonded via a water molecule to the conserved essential Glu490. The high resolution achieved also enables the placement of a significant number of water molecules and the identification of the electron transfer route between the heme $b$ and the binuclear site, the substrate access channel, and the ubiquinol binding site. Our structure of $Ax$qNOR demonstrates that high-resolution cryoEM structures of membrane metalloproteins are capable of defining key features that underpin catalysis involving redox sites.

## Results

### Overall organization of $Ax$qNOR and internal arrangement of redox centers

Structure determination was conducted on grids produced at the same time as those that generated the original cryoEM structure of $Ax$qNOR at 3.9 Å resolution in 2019[8]. However, through improvement in detector sensitivity (Falcon 4 camera), the use of a Selectris energy filter and more sophisticated data processing algorithms, we achieved a much higher resolution of 2.2 Å, placing it in the top 1% of cryoEM structures. Following extensive 2D and 3D classification, 400,000 particles were selected and reconstructed into this high-resolution cryoEM map of $Ax$qNOR (Fig. 1a, Supplementary Fig. 1, and Supplementary Table 1). This allowed for the placement of all sidechains within the cryoEM map for the measured Coulomb potential of the scattered atoms of a dimeric qNOR molecule and the tracing of a number of lipids, detergent molecules, several hundred water molecules, and a possible electron donor.

The overall fold revealed 18 α-helices packed together into each monomer (Fig. 1b), with residues 113–122, 227–255, and 564–571, creating the dimer interface. The heme $b$ and the binuclear $Fe_b$-heme $b_3$ centers are found within the core of the protein and are very well resolved within the map. The electron transfer heme (heme $b$) is linked to the catalytic core heme $b_3$ by a calcium ion to ensure their correct positioning during the functional cycle (Fig. 1c and Supplementary Fig. 2). The improved resolution (Fig. 1d) also permitted the identification of features not previously seen including well-ordered water molecules in the catalytic core, at the putative electron donor binding site and at the putative nitric oxide entry channel (Fig. 1b).

Overall the monomer arrangement of $Ax$qNOR (Supplementary Fig. 3a) is very similar to the arrangements of NorB, the catalytic domain of $Pac$NOR (Supplementary Fig. 3b) (rmsd for 390 Cα atoms is ~4 Å) and subunit N of $Ps$ $cbb_3$ cox (Supplementary Fig. 3c) (rmsd for 270 Cα atoms is ~8 Å). The main difference is in the number of helices. $Pac$NOR is composed of 12 transmembrane helices from the catalytic subunit B plus an additional helix from the heme $c$-containing subunit C. Heme $b$ is held by residues from TMII and TMX, while the Fe center and heme $b_3$ are held by residues from TMVI, TMVII, and TMX, respectively. $Ps$ $cbb_3$ cox contains 12 transmembrane helices in the central subunit N, one from cytochrome c-containing subunit O and two from subunit P (Supplementary Fig. 3). Heme $b$ is held by residues from TMII and TMX, Cu center and heme $b_3$ by TMVI, TMVII, and XII, respectively.

### Binuclear high-spin heme $b_3$ and Fe catalytic unit

The high-resolution cryoEM density map at the binuclear catalytic site located in the protein core has allowed for the accurate placing of the heme groups, as well as the associated metal ions, μ-oxo bridge, and newly identified water molecules linking to the conserved essential Glu490 (Fig. 2a and Supplementary Fig. 4). In addition to the oxygen of the μ-oxo bridge, two water molecules are found coordinating the heme group, with many more forming an extensive network in and out

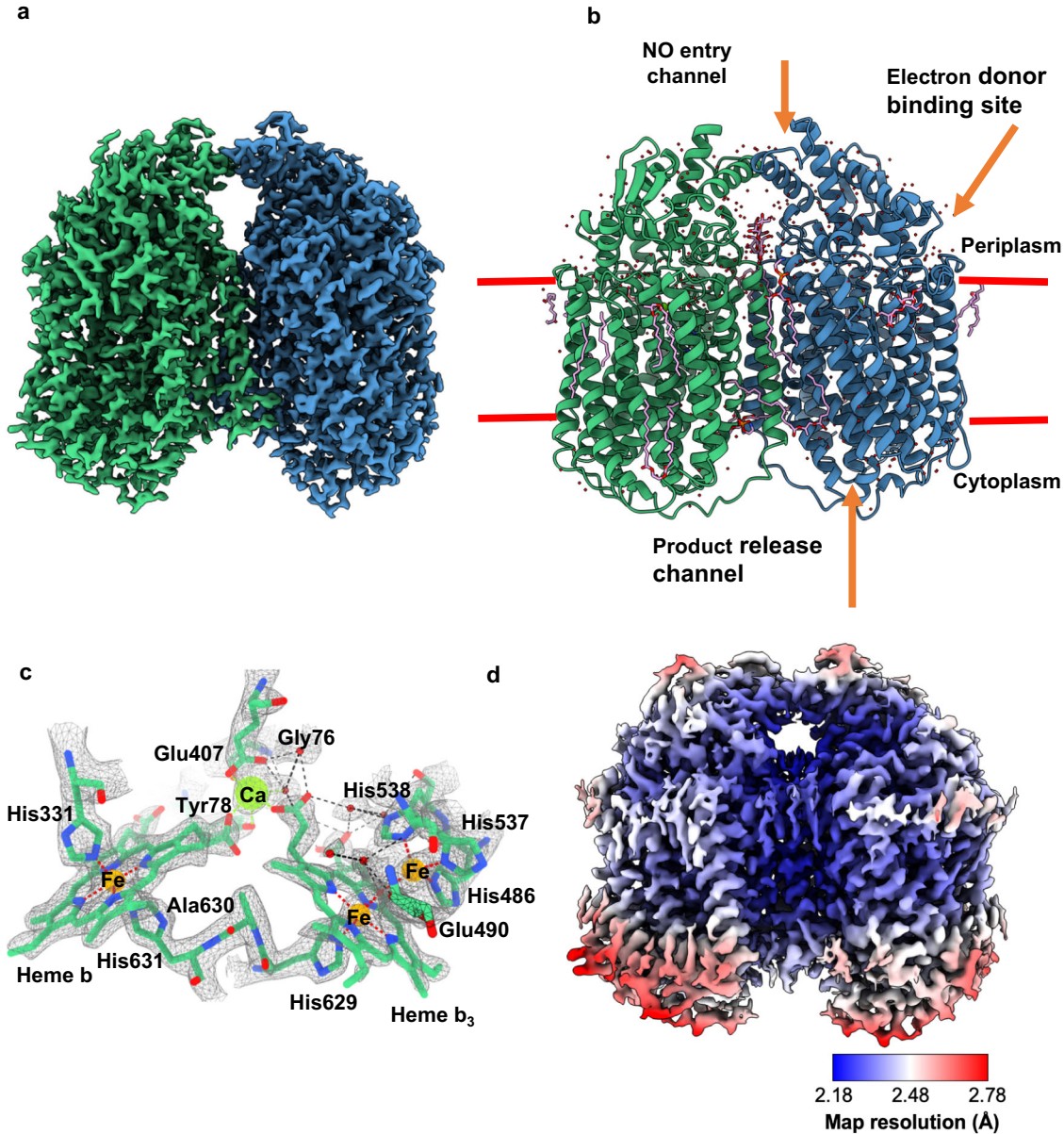

**Fig. 1 | The structure of *Ax*qNOR and its new features identified in a high-resolution cryoEM map. a** cryoEM map for the measured Coulomb potential of the scattered atoms of *Ax*qNOR determined by single-particle cryoEM to 2.2 Å resolution. Map colored by chain (A in green and B in blue). **b** Dimeric *Ax*qNOR in the plane of the lipid bilayer (red lines) colored by chain (A in green and B in blue), with waters colored as red spheres, DTM (decyl-thio-maltoside) detergent and LOP (lauryl oleyl phosphatidyl-ethanolamine (LOP) as magenta sticks. **c** *Ax*qNOR catalytic core formed of two adjacent heme groups linked by a calcium ion (green sphere) shown within the cryoEM density map. Iron ions are shown as orange spheres, water molecules as red spheres, and protein and heme groups as green sticks. All density maps contoured to a sigma level of 0.034 except for that corresponding to water molecules and Glu490, which is contoured to 0.020. **d** Map of *Ax*qNOR cryoEM density colored by local resolution.

of the active site. The Fe and water placement is organized primarily by six residues: Trp482, His537, His538, His486, Glu490, and His629, with the latter coordinating directly to $Fe_1$ in heme $b_3$. The role of these residues in forming the active site explains their sequence conservation in NORs and reinforces the evolutionary relationship of quinol oxidases/nitric oxide reductases within the heme-copper oxidoreductase superfamily[14].

Comparing the binuclear catalytic site in *Ax*qNOR with those in other family members identifies clear differences and striking similarities (Fig. 2). The μ-oxo bridge O connecting heme Fe and non-heme $Fe_B$ is clearly observed in *Ax*qNOR as well as in cNOR (Fig. 2a, e). Whereas in *Nm*qNOR, the distance from $Fe_B$ to the heme Fe does not permit the formation of a μ-oxo bridge (Fig. 2d). In the crystal structure of *Gs*qNOR, $Fe_B$ is substituted by a Zn ion which has likely come from

the chemicals in the purification and crystallization conditions (Fig. 2b). In this case also, Zn was too distant from heme Fe to form a bridging link, instead, it coordinates to two water molecules as well as three histidines, making it a penta-coordinate metal site. Though cNOR and *Ax*qNOR both possess a μ-oxo bridge between the heme $b_3$ Fe and $Fe_B$, the conserved Glu in the cNOR family (211 in *Pa*cNOR) is directly ligated to $Fe_B$, which is also the case for *Nm*qNOR, where Glu (494 in *Nm*qNOR) is also directly coordinated to $Fe_B$ despite the loss of the μ-oxo bridge. This variability in the conformation of this essential and fully conserved Glu is clearly not a difference between types of NOR (cNOR vs qNOR) but is most likely a manifestation of its genuine functionally relevant flexibility captured in different structures. Furthermore, this flexibility may help to accommodate two NO molecules in the tight binding pocket.

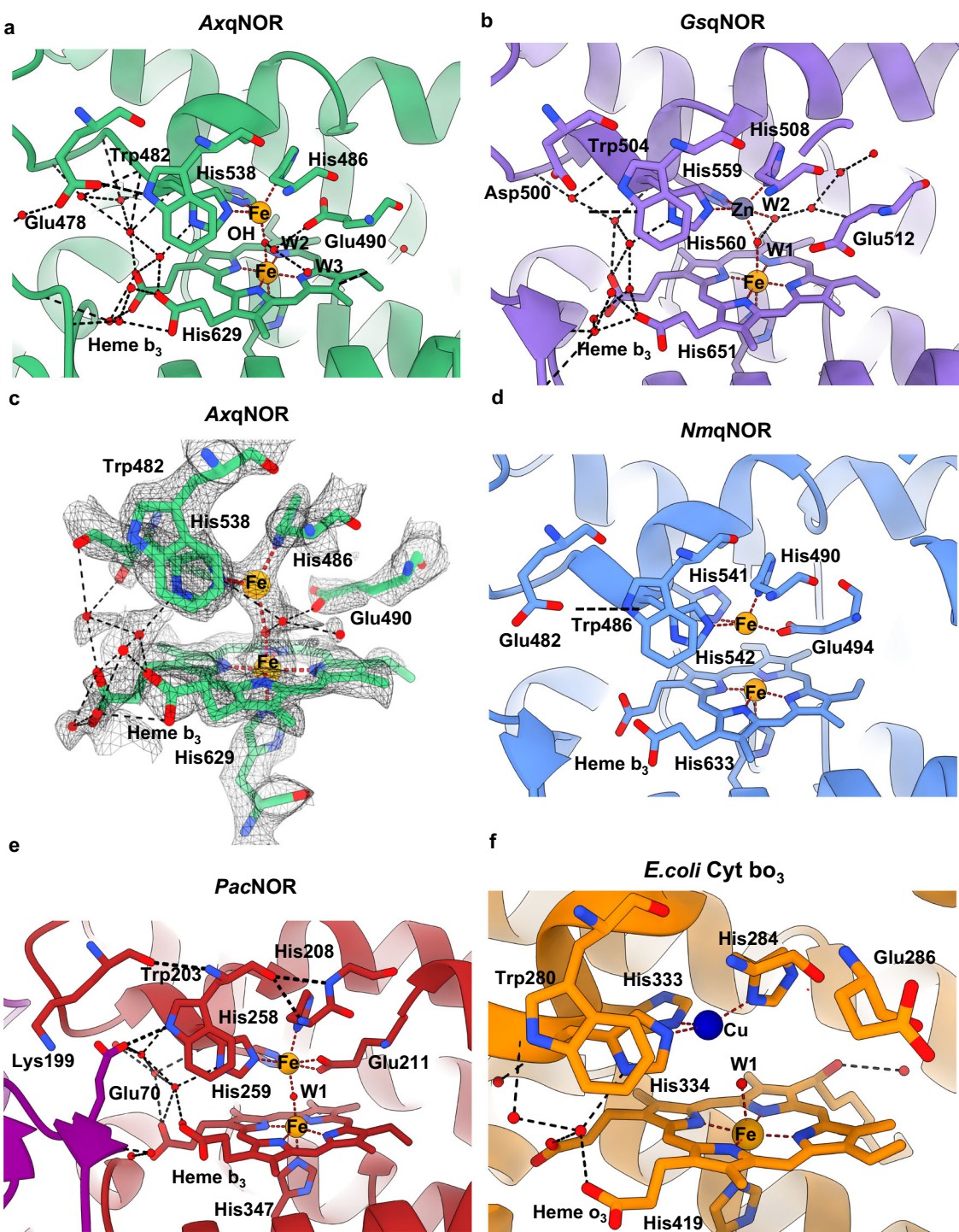

**Fig. 2 | Details of high-resolution AxqNOR and its comparison with other family members GsqNOR, NmqNOR, PacNOR, and E. coli cytochrome bo3.** a Binuclear center of AxqNOR with Fe_B coordinated by His486, His537, and His538 and μ-oxo bridge O. Iron atoms and water molecules shown as orange and red spheres, respectively. Hydrogen bonds and metal coordinating bonds are shown as black and red dashed lines, respectively. This scheme is used throughout the figure. **b** Binuclear center of GsqNOR (PDB ID: 3AYG) with a penta-coordinated zinc center. Glu512 is not bonded to ligands of the non-heme iron center. Zn is shown as a gray sphere. **c** Binuclear center of AxqNOR with the density map contoured to a sigma level of 0.034, except for that corresponding to Glu490 and water molecules which is contoured at 0.020. **d** Binuclear center of NmqNOR (PDB ID: 6L3H) with a non-heme Fe_B, which is coordinated by three histidine residues and Glu494. Fe ions are shown as orange spheres. **e** Binuclear center of PacNOR (PDB: 3O0R) with a penta-coordinated Fe_B center. Glu211 is bonded to a non-heme iron center. Water molecule W1 is bound to Fe_B and heme b3 Fe. **f** Binuclear center of E. coli cytochrome bo3 (PDB: 7N9Z) with a tri-coordinated Cu center. Cu is shown as a blue sphere.

The architecture of the binuclear active site is also preserved in quinol oxidases, as demonstrated recently by a 2.2 Å cryoEM structure of E. coli respiratory cytochrome bo3[21]. In this case, the heme Fe and Cu are 4.9 Å apart, not close enough to form a μ-oxo bridge. The distance from water W1 to Cu is 2.7 Å, while from heme Fe to W1, the distance is 2.3 Å. Although in close proximity, the conserved Glu (286 in this case) does not interact with the heme (Fig. 2f). In each family member, Fe in heme b3 is coordinated to a proximal His (1.9 Å) and connected to the

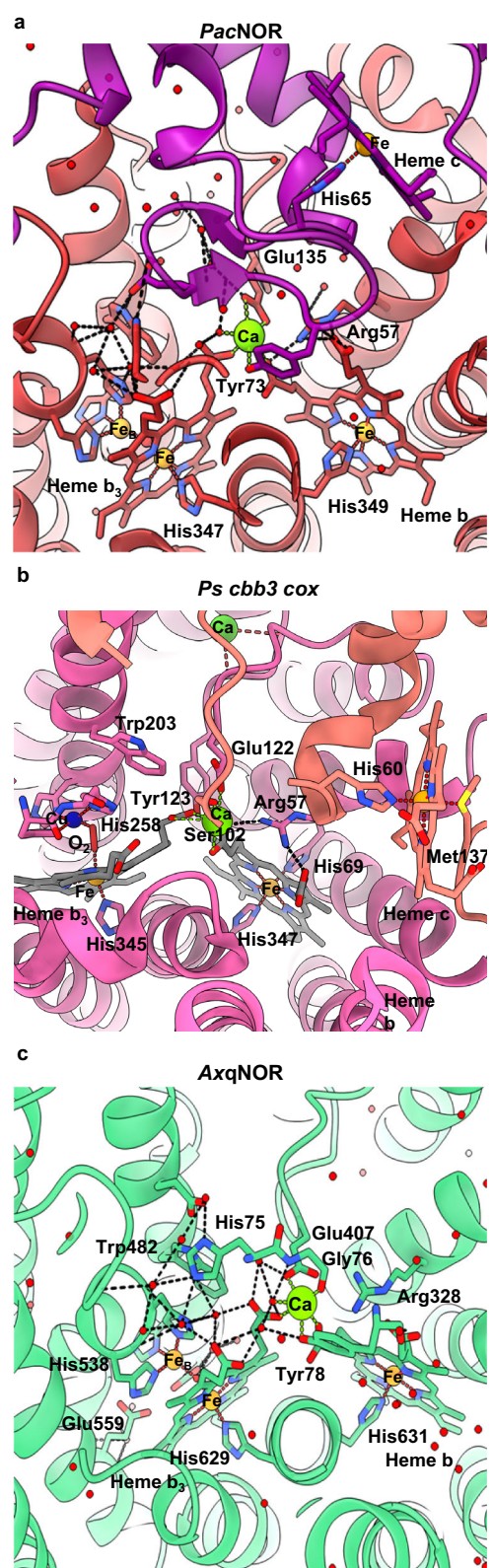

**a** *Pac*NOR

**b** *Ps cbb3 cox*

**c** *Ax*qNOR

**Fig. 3 | Comparison of *Ax*qNOR, *Pac*NOR, and *Ps cbb₃* cox calcium-binding sites. a** The *Pac*NOR Ca binding site is located on the interface between the CytC domain (purple) and the cNOR domain (red). Residues proposed to be involved in the electron transfer from heme c to heme b and heme bh are shown as sticks (PDB ID 3OOR). Water molecules are shown as red spheres, Fe atoms are shown as orange spheres, and calcium ions as green spheres. Hydrogen bonds are shown as black dashed lines and Ca and Fe coordination bonds are shown in green/red dotted lines, respectively. This scheme is used in the figure. **b** *Ps cbb₃ cox* Ca binding site (PDB: 5DJQ), Ca ion with hemisphere coordination is responsible for non-covalent fixation of ring D carboxylates of heme *b* and heme *b₃* from subunit N (pink) it also has a role of supporting subunit O (salmon) linker via Ser102-OD and Glu122-OE2. Cu atom is shown as a dark blue sphere, $O_2$ molecule, bound between heme *b₃* Fe and $Cu_B$ is shown as a red stick. **c** *Ax*qNOR Ca binding site, with protein shown as green ribbons and key residues highlighted as sticks.

## Role of calcium in maintaining communication between the electron donor, heme *b*, and catalytic unit

Ca ions have been identified to have multiple roles in metalloproteins, ranging from structural stability to conformational linkers of structural elements associated with regulatory function. In metalloenzymes specifically, calcium often connects different redox metals/clusters required for catalysis. For example, in N₂ORs, two metallo-clusters CuA and CuZ, in the electron-donating and catalytic site, are positioned in close proximity to each other by the Ca ion[22]. In Mn catalases, calcium ions are responsible for the compact hexameric arrangement[23] and in the photosynthetic center, Ca helps the active site to change its structure during the Kok cycle. In di-heme peroxidases and octaheme cytochrome c nitrite reductases, the Ca ion "bridges" two hemes via a hydrogen-bonded network of waters that connect to the propionate moiety of each heme[24]. Finally, in the cytochrome c nitrite reductase NrfHA, two of the four heme centers are bridged to a Ca ion by their propionates, which is the only route for inter-heme interaction[25].

In cNOR from *Pseudomonas aeruginosa* (*Pac*NOR), a $Ca^{2+}$ ion was identified linking heme *b* and heme *b₃* centers via their propionate moiety and was suggested to aid in arranging these cofactors for efficient electron transfer (Fig. 3a, b). In addition, the $Ca^{2+}$ ion is also coordinated by residues from heme *c*-containing subunits, such as Tyr73 in *Pac*NOR and Ser102 and Glu122 in *Ps cbb₃* cox. Our high-resolution structure of *Ax*qNOR confirms that qNORs also retain this arrangement of the propionates bridging the $Ca^{2+}$ site, and as such, are responsible for maintaining structural connectivity between heme *b* and heme *b₃* (Fig. 3c and Supplementary Fig. 5).

High-resolution *Ax*qNOR structure shows holospheric coordination of Ca ion, surrounded on all sides by oxygen atoms, different from previously determined structures (Supplementary Table 2). Calcium's preferred coordination numbers range from 6 to 8. In our structure, the Ca ion is ligated by seven oxygen atoms, OH Tyr78, O Gly76, OE2, and OE2 Glu407, single oxygens from propionates of heme *b* and heme *b₃* and a water molecule. The bond lengths are shorter than previously determined and now in the expected range of 2.1 to 2.6 Å, reflecting a greater accuracy of the structure. Unlike previous low-resolution structures of *Ax*qNOR and its mutant, O2D Heme *b* is too far to make a bond similar to *Pac*NOR. Water molecule was not visible in earlier determined structures due to limited resolution, Glu407 only contributed one bond, while propionate of heme *b* provides two bonds. Heme *b₃* and heme *b* are coordinated by His629 and His631, respectively, with both residues located on TM XII, and we postulate that electrons could be transferred through these sequential residues. In *Pac*NOR, the Ca ion is coordinated by equivalent residues Tyr73, Gly71, and Glu135 from NorC and propionates of heme *b* and heme *b₃* and a water molecule from NorB subunits (Supplementary Table 2). Its ligation by Tyr73 and Gly71 likely helps position the domain for effective electron transfer between heme *c* and the catalytic site via the His65-Ala72 route. Two histidines, His347 and His349 located on TMVII, coordinate heme *b₃* and heme *b* in similar positions to

proximal His of heme *b* (His631) via Gly630. Likewise, the non-heme metal, Fe in NORs and Cu in oxidases has a well-conserved tryptophan (Trp482 in *Ax*qNOR) in close proximity which interacts with Glu478, stacks against the non-heme Fe ligand His538 and appears to be at the end of a potential substrate access channel (see below). This comparison allows us to suggest that Trp482 may act as a substrate gatekeeper.

equivalent histidines in *Ax*qNOR (Fig. 3a, c). So, in the case of cNOR, the Ca ion likely plays additional roles in keeping the two domains together and in keeping heme *b* and heme *b₃* correctly poised for the efficient electron transfer required for catalysis in all NORs.

A close parallel can be drawn with the calcium-binding site in the *cbb₃*-type bacterial cytochrome *c* oxidases (*cbb₃*-oxidases), the second most abundant cytochrome *c* oxidase group after the mitochondrial-like *aa₃*-type cytochrome *c* oxidases. Like in NORs, the hemes *b* and *b₃* are linked together by a calcium ion, which is coordinated by the carboxyl groups of the pyrrole rings of both hemes[26]. The conservation of these structural features, together with significant sequence identities (Supplementary Fig. 6) between the NO reductases and the core complex of *cbb₃* oxidases, are consistent with the view that these reductases have a close evolutionary relationship[27].

## Electron pathway from quinol to the catalytic unit via Arg720, Asp724, and His305

In the map presented here, a weak density corresponding to a non-protein molecule is visible away from the transmembrane domain close to Asp724 and His305 (Fig. 4a). In the 2.7 Å crystallographic structure of *Gs*qNOR soaked with the quinol analog 2-heptyl hydroxyquinoline *N*-oxide (HQNO), a weak density was observed in a similar position[7]. This density was assigned to HQNO forming hydrogen bonds with Asp746 and His348 (*Gs*qNOR numbering) (Fig. 4b). We propose that the density in our map likely corresponds to an endogenous electron donor that was likely carried through from the *E. coli* in which the protein was expressed, since no such molecule or analog was added during sample preparation. Therefore, we have modeled the native bacterial electron donor ubiquinol into this density, which is accommodated well (Fig. 4a).

We then assessed its feasibility for being able to donate an electron to the heme *b* in *Ax*qNOR. His305, Asp724, and Thr721 form a pocket entrance which is compatible with this being the site of electron donor binding. Conserved among qNORs, Asp724 and His305 are H-bonded to ubiquinol (Fig. 4a and Supplementary Figs. 3a, 5). Residues Glu309 and Arg720 are hydrogen bonded to the ubiquinol ligands His305 and Asp724, respectively, with a critical Arg720 also making a stacking interaction with heme b, positioning it favorably for electron transfer. In addition, Arg717, Arg720, and Asn624 (RRN) form a triad of residues that create a water-filled cavity located close to heme *b*, making water-mediated electron transfer feasible[28, 29].

An important cluster of polar residues Arg71-Asp75-His98-Glu101 was first shown to form the ubiquinone binding site in cytochrome *bo₃*[30], a respiratory quinol oxidase. However, this cluster has not been fully identified so far from the sequence comparisons or in the available structures of NORs. The monomeric structure of *Gs*qNOR bound to HQNO revealed interactions with a number of polar residues (His328 and Asp746) and a hydrogen bond between Arg742 and the HQNO ligand, Asp746, in a manner also seen in *Ax*qNOR[7]. Our structure clearly shows that *Ax*qNOR not only contains the conserved His (305), Glu (309), and Asp (724) residues, but also possesses the critical Arg (Arg720) observed in cytochrome *bo₃*, which was suggested to be not present in qNORs[21]. Superposition of the electron donor onto the same site in *Nm*qNOR shows that *Nm*qNOR also contains this cluster (His303, Asp728, and Arg720) between the putative electron donor site and the redox center and that this cluster most likely fulfils the same role as proposed for *Ax*qNOR (Fig. 4). His305 and Asp724 and are well poised to become protonated upon the release of protons resulting from ubiquinol oxidation and to facilitate proton coupled electron transfer.

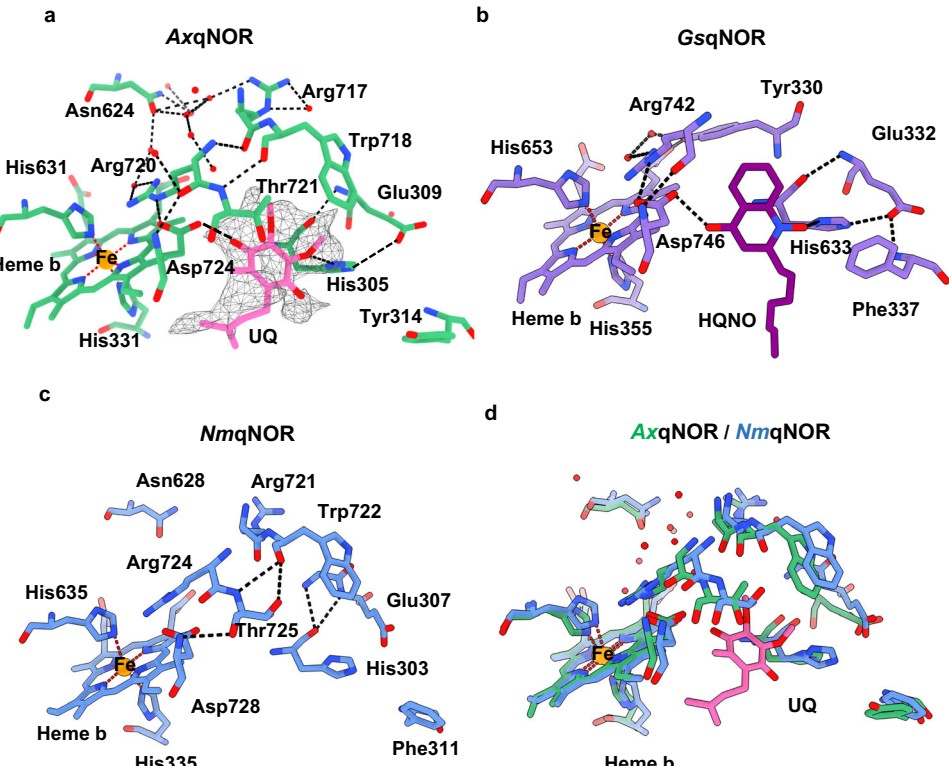

**Fig. 4 | Potential electron donor binding site identified in *Ax*qNOR map.**
**a** Proposed electron donor binding site in *Ax*qNOR showing weak cryoEM density for ubiquinol (UQ). *Ax*qNOR is colored green with residues lining the site highlighted as sticks, and the ubiquinol molecule is shown as pink sticks. The cryoEM density is contoured to a sigma level of 0.01 and shown as gray mesh. Water molecules and Fe ions are shown in red and orange spheres, respectively. This scheme is used throughout the figure. **b** HQNO binding site in *Gs*qNOR (PDB: 3AYG). *Gs*qNOR is colored in purple, with residues lining the binding site highlighted as sticks, HQNO molecule is colored in dark purple. **c** Potential ubiquinol binding site in *Nm*qNOR (PDB: 6L3H). *Nm*qNOR is colored in blue, with residues lining the binding site highlighted as sticks. **d** Superposition of *Ax*qNOR (green) and NmqNOR (blue) and the potential UQ binding site.

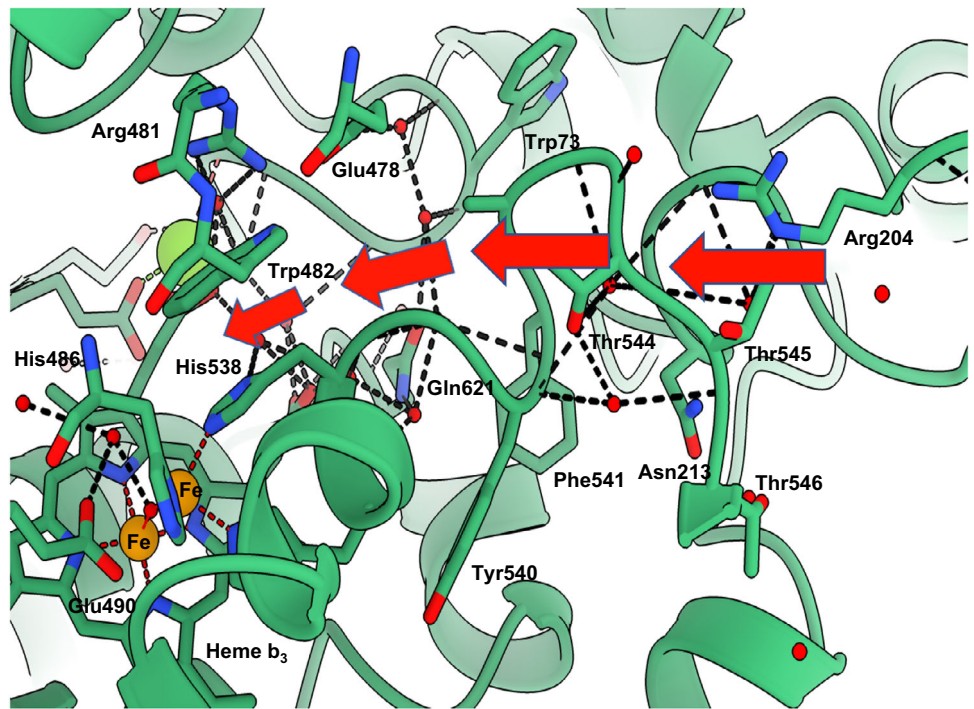

**Fig. 5 | *Axq*NOR putative NO path indicated by red arrows.** Cartoon representation of the molecule around putative NO path, indicated by red arrows. The path starts in the periplasmic part of *Axq*NOR and contains many charged and conserved residues. Important residues are shown as sticks, water molecules, iron, and Ca ions are represented by red, orange, and green spheres, respectively. Coordinating bonds are indicated by red dashed lines and hydrogen bonds by black lines. Labeled residues are fully conserved, except for Trp73 and Phe541.

## Substrate access, its delivery to NOR, and the role of Glu490 in NO binding

In denitrifying organisms, the fine-tuning of the levels of toxic NO has been suggested to involve product/substrate channeling from nitrite reductase (NiR) to NOR. Two genetically distinct types of nitrite reductase participate in denitrification, a cytochrome $cd_1$ type, encoded by NirS, and the more widely distributed copper-containing enzymes (CuNiR) encoded by NirK, as found in *A. xylosoxidans*. Both types of NiR have been shown to form a complex with c and q NORs[31] and the crystal structure of a 2:1 $(cd_1$NiR)–$(Pac$NOR$)_2$ complex from *P. aeruginosa* has been determined at 3.2 Å resolution. The authors consider that due to the topology of the organization of these enzymes in the membrane, and consistent with all-atom molecular dynamics simulations, a 1:1 complex formed by coulombic interactions of NiR with the membrane, is more likely formed in vivo[31]. The hydrophobic NO released from NiR preferentially diffuses into the membrane to bind to NOR, accessing the active site through the hydrophobic NO transfer channel identified in the *Pac*NOR structure. The retention of many structural features revealed from the crystal structures of qNORs and the catalytic domain of cNOR (see Supplementary Fig. 3a, b) suggest a similar scenario for NO channeling operates in *A. xylosoxidans*.

Based on early crystallographic structures, Val485 was considered important for guiding NO to the active site of *Axq*NOR. However, the mutation Val485Ala reduced the activity by only ~70%[8], suggesting additional determinants for NO entry. Computational analysis of NO diffusion using the *Pac*NOR structure has suggested the presence of at least one alternate path to the dominant migration pathway[6]. A similar study on cytochrome $ba_3$ had suggested the presence of hydrophobic tunnels located within the bilayer providing $O_2$ an area to partition from the aqueous phase, with little or no energetic barriers for $O_2$ transport[32–34].

Analysis of our cryoEM structure of *Axq*NOR and that of *Pac*NOR(7) has highlighted an attractive alternative route for NO entry, which is also compatible with binding the cognate partner protein CuNiR during turnover (Fig. 5, indicated by red arrows). This path, formed of fully conserved residues (Supplementary Fig. 6a), starts with the fully conserved Arg204. Located prior to TMI, this residue controls access to the entrance site for NO, and is followed by Asn213, along with multiple water molecules visible at the entrance of the tunnel. The water network is disrupted by the Phe541-Thr545 loop but continues afterward toward the loop close to the cavity created by Glu478, Gln621, and His75. His538, one of the non-heme Fe ligands, and Trp482 connected by stacking interaction, are located between this tunnel and the Fe site (Figs. 5 and 2a).

## Product release path and additional details of the water channel uncovered by the higher resolution structure

Details of the *Axq*NOR water channel are now visible in this higher resolution structure (Fig. 6a). For example, twenty water molecules are now identifiable in the density within the channel, with most being hydrogen bonded to polar residues and clusters of waters found within the cytoplasmic side of *Axq*NOR. Whereas, we only managed to identify three water molecules in this channel in the previously determined 3.3 Å resolution Val494Ala *Axq*NOR variant structure. There are also differences in the channel architecture. The conformation of Glu490, the gatekeeper of channel entry from the catalytic site, is different from the previously reported structure. Here, it is hydrogen bonded to the bridging hydroxyl and W2 (Fig. 2a), in contrast to being previously directed into the channel and hydrogen bonded with Glu559[9]. This high-resolution structure confirmed that Glu494 is flexible and has been modeled with half occupancy towards the channel and the other half directed away from the channel's opening. The Arg255-Glu572 salt bridge, visible in lower resolution structures, and located at the exit from the channel, is broken, and the sidechains of Arg255 and Trp251 are turned away from the channel and positioned to facilitate stacking interactions, widening the channel in doing so. In addition, the interactions between residues within this channel are slightly different for

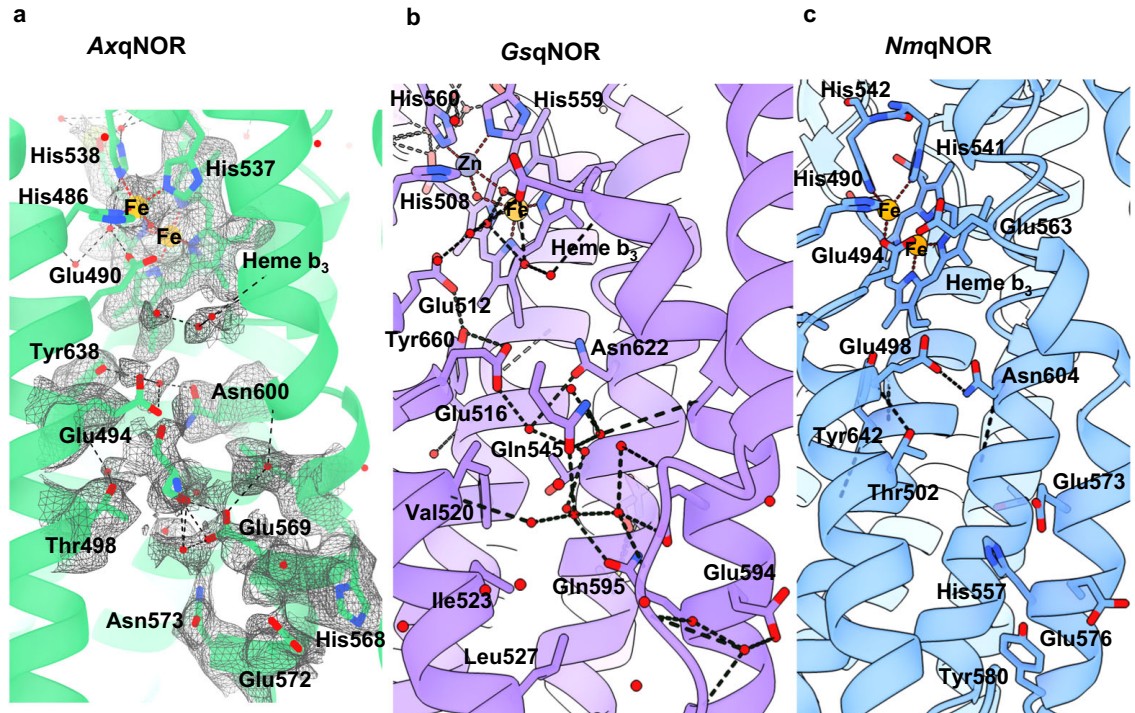

**Fig. 6 | Comparison of product release channels in structurally characterized qNORs. a** Product release path of qNOR from *Alcaligenes xylosoxidans* (*Ax*qNOR) from the binuclear center (top) to the cytoplasmic side (bottom). *Ax*qNOR colored in green with residues lining the proposed path in *Ax*qNOR shown as sticks. Fe atoms and water molecules are shown as orange and red spheres, respectively. Hydrogen bonds and metal coordination bonds are shown as black and red dashed lines, respectively. This scheme is used throughout the figure. **b** Product release channel in qNOR from *Geobacillus stearothermophilus* (*Gs*qNOR) (PDB ID: 3AYG) colored in purple, with residues lining its proposed proton channel highlighted as sticks. Zn is shown as a gray sphere. **c** Product release channel in qNOR from *Neisseria meningitidis* (*Nm*qNOR) colored in blue, with residues lining its proposed proton channel highlighted as sticks (PDB ID: 6L3H).

qNORs from different organisms. In *Nm*qNOR, Asn604 and Glu498 make a strong hydrogen bond (Oe2 Glu498 to Nd2 Asn604 distance is 2.5 Å), while in *Ax*qNOR and *Gs*qNOR, their analogous residues are connected via binding waters (Fig. 6). It has been shown earlier by molecular dynamics simulations of *Gs*qNOR that bulk solvent in a similar channel is freely exchanging water with the bulk water of the cytoplasm[7], suggesting this functions as a proton delivery pathway.

The Phe253- Pro262 loop displays a different conformation to that shown in previously determined lower resolution structures of monomeric qNORs. In *Gs*qNOR, this loop opens a different path directed towards the dimer interface (Supplementary Fig. 2). In *Ax*qNOR, this path is blocked by a lipid molecule positioned between this interface (See Fig. 7c).

We conducted a search for additional potential channels that may serve for product release from the active site, starting the search from a binuclear site using CAVER[35]. This analysis identified a long hydrophobic channel (Supplementary Fig. 7) directed into the cytoplasmic area. It originates from the binuclear site cavity, composed of multiple hydrophobic residues (Supplementary Fig. 7b). It follows around helix VII and opens up into the cytoplasm. Helix IV serves as a boundary between this channel and the water channel (Supplementary Fig. 7). Majority of the residues lining the path are strictly conserved (Phe337; Leu344; Phe349; Leu350; Val367; Leu370; Leu374, Trp488, Val489; Phe496; Phe503; except for small number of semiconserved residues (Leu336; Ile354; Ile423; Phe466; Val649) shown in the Supplementary Fig. 7b.

### Presence of lipids and detergent molecules in the *Ax*qNOR map
In addition to the identification of water molecules within the map, several areas of additional density were observed, including the long tubular density (Fig. 7a), which is consistent with that observed for bound lipids with classical "U" shaped density. They can be clearly seen positioned within the transmembrane region of the protein (Fig. 7b). However, no additional lipids were re-introduced during protein purification, so these densities represent lipids carried through the extraction and purification process. In our structure, they have been modeled as the native *E. coli* lipid Lauryl Oleyl Phosphatidylethanolamine (LOP). As is common with detergents and lipids, the density is not well defined for the tail regions, likely indicative of its flexibility. In our model, where lipid density is weak, the sections of lipid not covered by density have not been modeled in. The lipids make interactions with the exterior of the transmembrane helices but do not appear to make any specific interactions with the substrate tunnels or the bound cofactors.

Additional density at the dimeric interface is not consistent with the density expected for lipids. Instead, the strong head density and single tail is more typical of a bound DTM detergent molecule used during *Ax*qNOR extraction and purification (Fig. 7d). Its location at the dimeric interface suggests that DTM replaced a bound lipid molecule during protein extraction. The presence of the lipids within the dimeric structure further reinforces that the integrity of the structure of functional dimeric structure is well preserved in this cryoEM structure.

## Discussion
Many enzymes exploit the oxidation states of metals to perform redox cycling during catalysis. In many metalloenzymes, catalysis involves the controlled delivery of electrons and protons to the active site where chemical substrates are utilized. These events are often coordinated, coupled, and orchestrated by structural signals that remain poorly understood in many cases due to the experimental limitations, particularly for membrane proteins, that require solubilization and can be difficult to crystallize. Consequently, although the number of

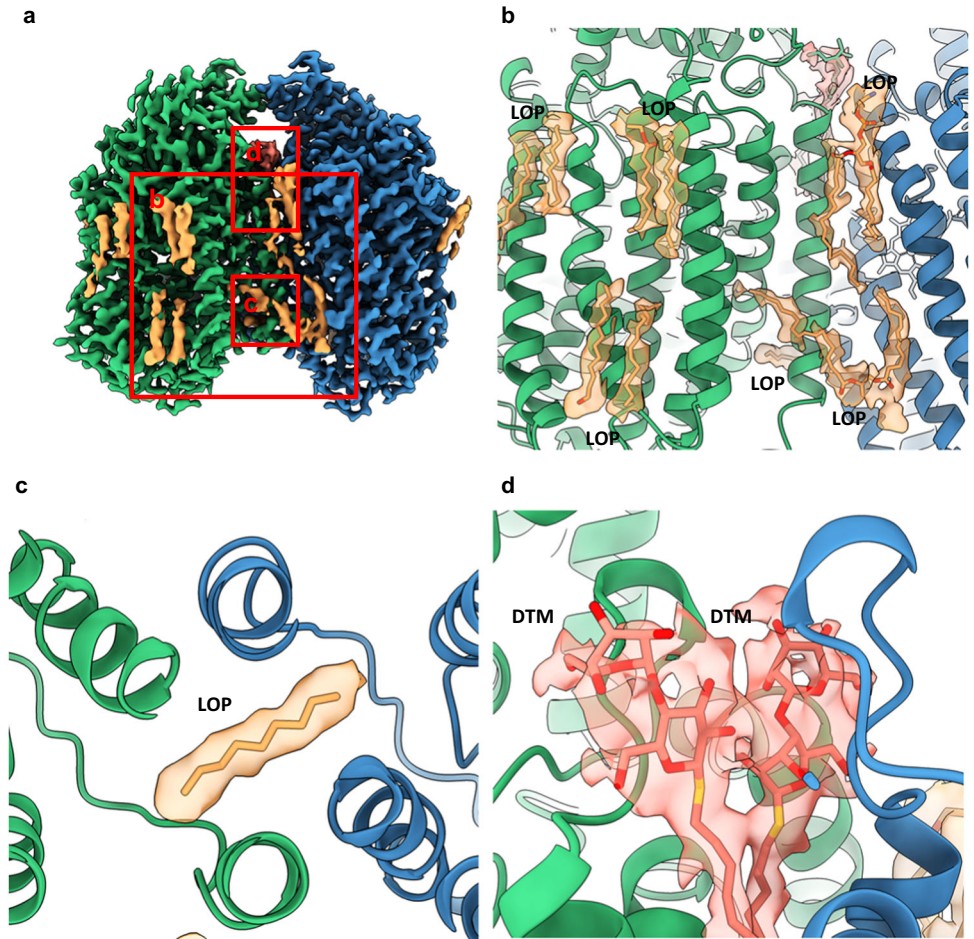

**Fig. 7 | Ordered lipids and detergents present in the cryoEM map of *Ax*qNOR.**
**a** Overview of additional non-protein density in the *Ax*qNOR map. Chain A density is shown in green, chain B density is shown in blue, LOP lipid density is shown in orange, and DTM detergent is shown in salmon. The density for protein is contoured to a sigma level of 0.030 and the density for detergent and lipids is contoured to a sigma level of 0.008. This scheme is used throughout the figure. **b** Modeled LOP lipids found at the protein exterior and dimer interface. **c** Modeled fragment of LOP crossing the dimer interface and protruding into the product release channel. **d** DTM detergent at the dimer interface.

unique structures for membrane proteins have steadily increased since the first structure of a membrane protein in 1984[36], progress has been slower than predicted. However, advances in cryoEM has provided a major boost to the structure determination of membrane proteins, including membrane metalloproteins[11,37], but only around one percent of the structures are determined to high enough resolutions (~2 Å) required to understand the catalytic mechanism of metallo-membrane proteins.

Nitric oxide and oxygen reductases that form the respiratory heme-copper oxidase superfamily (HCuO) have been the focus of study for many years due to their importance in bacterial respiration. Recently a 2.19 Å cryoEM structure of *E. coli* respiratory cytochrome *bo₃* revealed bound phospholipids and ubiquinone-8 in a dynamic substrate binding site[21]. The 2.2 Å structure of *Ax*qNOR presented here is the highest resolution structure of any NOR (cytochrome or quinol-dependent) by any method, allowing a comparison of these distinct members of the superfamily, one involved in $O_2$ respiration and the other in nitrate respiration. The high-resolution cryoEM structure of *Ax*qNOR clearly establishes the presence of a μ-oxo bridged binuclear site similar to the cytochrome-dependent NOR. This conclusively demonstrates that this is not a genuine difference between a quinol-dependent and cytochrome-dependent NOR. The separation between the heme Fe and the non-heme metal ($Fe_B$ in NORs and $Cu_B$ in cytochrome oxidases) shows variability. This, together with the variability in the conformation of the essential and fully conserved Glu (490 in

*Ax*qNOR, 494 in *Nm*qNOR, and 211 in *Pac*NOR), clearly reinforces the importance of these structural features in the binding of NO and $O_2$. These structures have provided a significant advance allowing the identification of structural factors that may be involved in (a) stability of functional dimer, (b) binding of the quinol electron donor, (c) protein–protein interaction, (d) binding of NO and provision of electron and protons for NO catalysis, and (e) a pathway for the release of the product. These structurally informed putative routes can be probed and tested for qNOR in combination with functional studies to establish factors that are important for molecular recognition, specificity, and catalysis with wider implications to many redox enzyme systems, including those of the respiratory heme-copper oxidase (HCO) superfamily. We note that bacterial oxidases of the HCO family, including the *caa₃*-HCO and *ba₃*-HCO from *T. thermophilus*, *cbb₃*-HCO from *P. stutzeri*, and *bo₃*-HCO from *E. coli*, are able to catalyze the reduction of NO to $N_2O$[37–39].

## Methods

### Purification of recombinant *Ax*qNOR and sample vitrification

*Ax*qNOR protein production and vitrification on a cryoEM grid followed the method stated in Gopalasingam et al.[9]. Briefly, His-tagged *Ax*qNOR-BRIL was expressed in *E. coli* cells and the protein extracted from *E. coli* membranes in a buffer containing 1% *n*-Dodecyl-β-ᴅ-maltoside (DDM). The protein was then purified using Nickel affinity chromatography, then size exclusion chromatography in a Tris-HCl

buffer pH 7.0, 150 mM NaCl, and 0.05% decyl-thio-maltoside (DTM), before concentrating for storage. When making grids, a sample at 3 mg/mL was applied to glow-discharged Quantifoil Au R1.2/1.3 grids and plunge frozen using a Vitrobot Mark IV (FEI).

## CryoEM data acquisition

All cryoEM data were collected on an FEI Titan Krios TEM at the Astbury Biostructure Laboratory at the University of Leeds. The 300 kV microscope was equipped with a Falcon 4 camera and a selectris energy filter set to 10 e⁻ width. In total, 5466 images were taken at a nominal magnification of 130,000x, a pixel size of 0.91 Å/pixel, and a defocus range was used of −0.9 to −2.7 μm. A total dose of 34.90 $e^-/Å^2$ was applied to the movies over an exposure time of 6.11 s, corresponding to a dose per $Å^2$/second of 5.71. The movies were split into 34 frames, giving a dose per frame of 0.8 $e^-/Å^2$.

## Image processing of cryoEM movies

For the *Ax*qNOR structure, all image processing was performed in RELION 3.1[40] unless otherwise stated. For pre-processing, beam-induced motion correction was performed on the movies using RELION's own implementation MotionCor2[41], then the contrast transfer function (CTF) of each micrograph was estimated and corrected using CTFFIND-4.1[42,43]. The particles were picked on the motion-corrected micrographs by crYOLO 1.6.1 (Sphire)[44] using the weights from its general model and a picking threshold of 0.1. In total, ~3 million particles were picked and extracted from the CTF-corrected micrographs and put through two rounds of 2D classification. After removing poorly aligned particles and those that belong to poorly resolved or "bad" classes, 1.8 million particles remained. These particles then underwent 3D classification using the previously solved *Ax*qNOR structure (PDB: 6QQ5)[9] as a 3D reference filtered to 60 Å resolution to avoid model bias.

The resulting 730k particles were refined using RELION's auto-refine program, then underwent a further round of 3D classification, leaving 404,950 particles. The unbinned particles were refined in C2 symmetry to give a resolution of 3.77 Å. The refined map was post-processed in RELION, whereby it was sharpened according to the global B-factor with a mask applied around the protein region, which improved the resolution to 3.31 Å. After two rounds of both Bayesian Polishing and per-particle CTF refinement, the resolution had improved further to 2.73 Å. In order to better model the de-localized CTF signals, and to follow the guidance that smaller proteins require larger box sizes[45], the particles were re-extracted with a larger box size. The particles underwent three further rounds of both Bayesian Polishing and CTF refinement. After refinement and then masking out the solvent and detergent micelle, the final resolution as calculated by the gold standard half map criteria with a 0.143 cut-off, was 2.247 Å.

## Model building and refinement of cryoEM structures

The structure was solved by molecular replacement using a starting model of *Ax*qNoR (PDB ID: 6QQ5) and the program MOLREP[46] of the CCPEM suite[47]. This was followed by jelly-body refinement in REFMAC5[48] in CCPEM against the half maps, and restrained refinement with local NCS to final Fcs average of 0.85 and RMSD 0.012 Å.

The *Ax*qNOR model (PDB ID: 6QQ5) was initially built into the post-processed map that was sharpened according to the global B-factor. Density modification[49] was performed in Phenix 1.20 using the final FSc half maps and a refined model of *Ax*qNOR. This gave a map with improved density, so further model building was performed on this map, and then refinement was performed using the non-sharpened, non-modified map, followed by manual rebuilding in Coot[50]. Model building was assisted using cryoEM density difference maps produced by Servalcat in the CCPEM suite[51]. Maps were visualized and figures were produced using ChimeraX[52].

## Sequence alignment

Sequences were first aligned by pairwise alignment against the *Ax*-NOR sequence using BLAST and scored according to the BLOSUM62 score matrix. Any sequences not similar enough to Axq-NOR for pairwise alignment were aligned by multiple sequence alignment with the upper sequences using Clustal Omega. These aligned sequences were not scored by a matrix, and so were marked only for identical residues.

## Reporting summary

Further information on research design is available in the Nature Portfolio Reporting Summary linked to this article.

## Data availability

All data needed to evaluate the conclusions in the paper are present in the paper and/or the Supplementary Materials. The CryoEM data generated in this study have been deposited in the PDB database under accession code 8BGW, CryoEM map has been deposited in the EMDB database under accession code EMD-16041. Previously solved protein structures used in this manuscript are available from the PDB. These structures are as follows: 6QQ5 (*Ax*qNOR), 3AYG (*Gs*qNOR), 6L3H (NmqNOR), 3OOR (PacNOR) 5DJQ (*Ps cbb₃ cox*), and 7N9Z (*E. coli* cytochrome bo3).

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

## Acknowledgements

Samples and grids were prepared by Chai Gopalasingam and Rachel Johnson, respectively, for the original publication (ref. 8). Grids stored at the time were used for fresh data collection at Astbury Biostructure Laboratory at the University of Leeds, with assistance from Louie Aspi-nall. Richard Strange helped with the installation of CAVER software and analysis. The work on denitrifying enzymes in Liverpool was supported by BBSRC (BB/L006960/1 and BB/N013972/1). A.J.F. thanks the Well-come Trust for their PhD studentship funding. The Titan Krios was fun-ded by the University of Leeds and Wellcome Trust (108466/Z/15/Z) and the Falcon 4 detector and selectris energy filter were funded by Well-come Trust (221524/Z/20/Z).

## Author contributions

A.J.F. set up the data collection and processed the cryoEM images. S.V.A. built and refined the models. S.P.M., S.V.A., and S.S.H. conceived the study. All authors (A.J.F., S.V.A., R.R.E., S.P.M., and S.S.H.) contributed to the writing of the manuscript, analysing the data, and interpretation of results.

## Competing interests

The authors declare no competing interests.
