## [Peer review file · Nature Communications]

REVIEWER COMMENTS

Reviewer #1 (Remarks to the Author):

The paper is an extension of the authors' previous work. They had previously determined the structure of a quinol dependent NO reductase at 3.2 Å. They have now improved the resolution to 2.2 Å. Here they present novel findings and discuss them in greater detail. However, the paper is written in a careless manner, and I would send it back, at least in order to use a spell checker.

Although the amount of novel information is limited, it could be published after a major revision. Novel information comprises the structure of the active site of qNORs, but the paper lives mainly from comparisons and reinforcements. The points to be addressed are listed below:

The reader would be interested to learn what led to the improvement in resolution. Please provide the information.

The calcium binding site is nearly identical to the one in the C-family (cbb3 oxidases) of the HCOs. These are most closely related to the NO reductases. The Ca binding site therefore is an issue of conservation. Mention and discuss.

p.3 Why should a dimer be the active species? Is there any mechanistic reason? Please discuss!

p.4 There should be a ubiquinol binding site, but not a ubiquinone binding site. Binding of ubiquinone is detrimental because it would lead to substrate inhibition.

p.5 Figure 1 legend: In EM you do not get „electron densities“, electron densities are obtained by X-ray crystallography. Introduce the abbreviations DTM and LOP

p.6 correct reductases, quinole, have to have, add „water“ before W1

p.7 correct comparison, Figure 2f: replace bo3 by bO3, Cu is shown in orange not in blue, replace Cytochrome bo3 by cytochrome bO3, correct cytochrome

p.8 correct Calcium, pseudomonas,

p.9 correct correctly, the electron donor is ubiquinol, not ubiquinone. The role of asp124 and his305 could be to take over the two protons from the ubiquinol which are released upon ubiquinol oxidation.

p.10 Figure 4C,D, I am confused: Why should electron transfer from the quinol to the heme b iron require a „pore“ ???

p.11 introduce CuNiR, not mentioned before. What is its function, its relation to the qNOR? Correct channel

p.13 figure legend correct channel, Lauryl, Oleyl, indictive

p.14 correct transmembrane, remove of the structure, change has to have

p.15 correct separation, introduce DDM abbreviation,

p.16 correct QNOR

Other issues:

- C-terminus, and not C-terminal

- No author contribution for Robert E. Eady

- The correct term is electron cryomicroscopy, not cryo electron microscopy, because the electrons are not cryo, but the microscopy is done at cryo temperatures

Reviewer #2 (Remarks to the Author):

The manuscript by Flynn et al reports on a 2.2 Å resolution structure of the qNOR from *Alcaligenes xylooxidans* (Ax). The resolution is undoubtedly better than the (up to) 3.2 Å resolution structure of the same protein reported previously by the same group, but I do not find the quality of the analysis sufficient to warrant publication of the manuscript in its present form. I would like to add that the previous publications on the same or closely related proteins from the same group (their references 8 and 9) were clearly written, data well supported and conclusions reasonable and interesting, so it is unclear to me why the current manuscript lacks the same qualities.

My major general points are that it is unclear what exactly the higher resolution leads to in terms of new insights and that the analysis made is unclear and not supported by any complementary methods such as computational analysis, sequence analysis or mutational analysis. Many points made are unclear at best and at worst show misunderstanding about the heme-copper oxidase superfamily. The specific points made below are included to explain my concerns. It should also be noted that I cannot evaluate the technical quality of the cryo-EM analysis, which is hopefully covered by other reviewers.

Specific comments:

1. At the top of p.6 sequence conservation and evolutionary relationships to the heme-copper oxidases (HCOs) are described in a very unclear way, and no sequence alignments are included. What is conserved where? The Glu-490 mentioned is e.g. NOT conserved to the oxygen-reducing HCOs.

2. p. 6. Last paragraph: The comparison to the E.coli bo3 O₂-reducing HCO is very unclear. Why focus on this here? The comparison when comes to quinol oxidation/binding is relevant but the bo3 active site is no more similar to qNORs than other members of the O₂ reducing HCOs are. Why would you expect a μ -oxo-bridge there? It has never been observed in any of the many structurally characterized O₂-reducing HCOs. Also the sentence 'the conserved Glu (286 in this case)' is misleading since this is NOT the same Glu as in the NORs. This Glu (286 in bo3) is conserved to the A1-type HCOs only (sequence motif HPEVY), not to the B or C-type HCOs and NOT to the NORs (where the Glu mentioned is rather part of the different conserved HLWVE motif). The Glu mentioned in bo3 (286) sits at the end of a proton transfer pathway (D-pathway) to shuttle protons to the active site (and to be pumped) and fulfills a different role from the Glu (494 in Nm qNOR) in the NORs which is a ligand (at least during parts of the catalytic cycle) to the FeB. Why are these compared in this way? Sequence alignments, if supplied, would clearly show this. And the same goes for the 'well-conserved Trp482' interacting with Glu-468. Here Glu-468 is not shown in the figure 2, but only much later (Fig 5). This should be clarified, sequence alignments provided (or at least clear reference to previous sequence alignments made) and Uniprot identifiers given as these sequences are not easy to find. Also, 'in each family member, Fe in heme b3..' is a bit misleading, as the heme is a heme o in the bo3.

3. The Ca²⁺ discussion on p8 is unclear; why is this discussed here? What does the higher resolution provide that was not observed before in the same protein (ref 8)? The Ca site was identified in the previous structure already. Also, the C-type HCOs show similarity to the NORs in many respects such as the presence of a Ca ion in structurally the same position (see PDB ID 3MK7), but this is not mentioned in the list of other, much less related, Ca-containing metal proteins.

Further on the same topic is stated 'We postulate that electrons could be transferred through these residues (H629/H631)..'. Why? Electrons can tunnel, what is the basis for suggesting the transfer through specific residues at all in general and through these ones in particular? No support for this claim is presented. The continuation 'rather than solely through the Ca ion as suggested for cNOR' is even more unclear. I cannot find that this has ever been suggested for cNOR, the Ca is suggested to stabilize the hemes for efficient transfer directly between hemes (NOT through calcium which to my knowledge has never been observed to be redox-active). This section would need major revision.

4. The quinol site discussion on p9-10 is in principle interesting. However the text does not correspond to the figure, modelling of the quinol is not clear (there is no picture with the model overlaid with the actual density so that the quality can be evaluated). And what would the native quinol donor be in Ax? Also the His305, Asp724 and Thr721 are said to 'form a neat pocket entrance', but apart from Asp-721 the fig 4B does not show how this would be accomplished, what is actually suggested to interact with the UQ molecule forming its binding site? (H305 I cannot see at all...). Also a RRY triad is suggested to interact with H₂O to make 'water-mediated electron transfer' feasible. Where are the references for such 'water-mediated e⁻'? Electrons generally prefer to tunnel through hydrophobic interiors where the reorganisation energy is low. Further, the discussion on the residues His305, Asp724 and Arg720 is also unclear; which ones are conserved also to the O₂-reducing quinol oxidising HCOs? And which ones to qNORs? Again, sequence alignments would clearly show this. If there are residues that occupy the same spatial location as in the O₂ reducing (quinol oxidising) HCOs but at a different (conserved) sequence location in qNORs, this would be interesting.

5. The NO path discussion on p 10-11 is very unclear. On the top of p. 11 a possible binding to a NIR partner protein is mentioned without any background or reference to where and if this has been observed previously. The alternative route suggestion for NO is made without any support from calculations/tunnel analysis and is further suggested to contain water and charged and/or polar molecules, whereas all previous work in the area searched for hydrophobic pores/tunnels to accommodate gases like NO or O₂. These types of suggestions would need to be supported by analysis of sequence conservation and by feasibility in terms of computational and or experimental support.

6. The product release path discussion on p 11-12 is also unclear. The first question is which product is discussed? Water or N₂O? There is then discussion of 'the' channel or pore without any introduction to what I presume is the water channel observed in previous qNOR structures suggested to be used for proton uptake. Here there is no mention/analysis of the possibility of proton uptake through this channel and no interpretation of what the differences to other structures would imply for the function. It is also not clear which details are better observed in this higher-resolution structure compared to the same authors' previous work.

7. The concluding remarks on p 14-15. Most of the text in this paragraph is rather a general introduction than a conclusion of the most important findings.

Minor points

1. p5, top lists new features observed and refers to Fig 1B, where I can see no indication of 'putative nitric oxide entry channel'. Rephrase or show.
2. p2, the sentence that starts : 'Three types of respiratory bacterial NOR are recognized' is unclear as it sounds as if the three types that are then mentioned belong to these three categories (which they don't); rephrase.
3. p3, middle paragraph: 'These families have ahigh-spin heme b₃'. Not correct, heme-copper oxidases can have heme a₃, heme b₃ or heme o₃ in the active site.
4. p.6 middle paragraph refers to cNOR for Fig 2c, but it depicts GsqNOR.
5. Many places states 'Ca atom', 'Zn atom' etc when should be 'ion'.
6. There are many spelling mistakes, typos and incorrect use of italics which hinders the 'flow' of reading. A simple run through a spell-checker should help.

Reviewer #3 (Remarks to the Author):

Review comments

The article “A 2.2Å cryoEM structure of a quinol-dependent Nitric Oxide Reductase shows close similarity to the respiratory quinol oxidase family” by Alex J. Flynn et al. seeks to address how Quinol-dependent Nitric Oxide Reductases (qNORs) bind to quinol, and how they play a role in electron transfer. The 2.2Å cryo-EM structure of qNOR from *Alcaligenes xylosoxidans* provided informative details of the metalloproteins that are capable of defining key features that underpin catalysis involving redox sites.

In general, the article’s results are solid, the conclusions are clear, and the discussions shed important light on respiratory heme-copper oxidase superfamily. I would be favorable for the publication of this manuscript. Some of their conclusions, statements or speculations may need citations of previous results to support their current analysis. Some figures may also need to be reorganized and recolored to highlight the contents. Given these potential shortcomings, it will be appreciated if this manuscript can be carefully revised before publication.

Major points:

Figure 1. B did not show the putative electron donor binding site and the nitric oxide entry channel. Although the water molecules could be seen, it’s hard to define the critical waters.

Figure 1. C, it is hard to find the Glu490. The author could either enhance the color or change the color to highlight this key residue—there are similar problems among all figures.

“The μ -oxo bridge and newly identified water molecules linking to the conserved essential Glu490.” This glutamic acid seems conserved among other family members. Providing a sequence alignment will be highly appreciated. It would be better if the authors can design point mutations to verify the water molecule linking function. Same for Trp482.

Figure 3. C aligns the C alpha atoms by showing the similarity. Quantitative analysis for alpha distance is needed.

Figure 4. A did not show Asp724 and His305, which have been mentioned in the main paragraph. Figure 4.D did not show the superimposition between NmQNO and AxqNOR.

It is nice to obtain an alternative route for substrate access based on the high-resolution cryo-EM structure of AxqNOR. If possible, I suggest the authors have biochemical or computational evidence to verify these results.

“DTM replaced a bound lipid molecule during protein extraction”, but in the method part, the authors mentioned that the AxqNOR was extracted with 1% DDM. Which one is right?

Extracted the AxqNOR with 1% DDM but purified with 0.05% DDM. Why choose DDM? In the 2021 paper, they used styrene–maleic acid co-polymer (SMA) nanodiscs and membrane scaffold protein (MSP) nanodiscs. Why did the authors choose different detergents? And does the DDM/DDM help to get the highest structure of NOR?

Minor points:

Reference is needed to support “a weak density was observed in a similar position”.

Reference is needed to support “An important RRY triad was first discovered in cytochrome bo3...”

Reference is needed to support “ Based on early crystallographic structures, Val 485 was considered important for guiding NO...” and “mutation of Val485Ala only reduced the activity only by 60%...”. Check whether it has a redundancy “only”.

“They can be clearly seen positioned within the transmembrane region of protein...” In this sentence, “clearly ” should be “clearly”.

REVIEWER COMMENTS

Reviewer #1 (Remarks to the Author):

The paper is an extension of the authors' previous work. They had previously determined the structure of a quinol dependent NO reductase at 3.2 Å. They have now improved the resolution to 2.2. Here they present novel findings and discuss them in greater detail. However, the paper is written in a careless manner, and I would send it back, at least in order to use a spell checker.

Although the amount of novel information is limited, it could be published after a major revision. Novel information comprises the structure of the active site of qNORs, but the paper lives mainly from comparisons and reinforcements. The points to be addressed are listed below:

We thank the reviewer for thoroughly reviewing the manuscript helping to improve it significantly.

1. The reader would be interested to learn what led to the improvement in resolution. Please provide the information.

We have added a paragraph at the end of introduction and the opening paragraph of the Results section highlighting the features that has led to the improvement in resolution.

2. The calcium binding site is nearly identical to the one in the C-family (cbb3 oxidases) of the HCOs. These are most closely related to the NO reductases. The Ca binding site therefore is an issue of conservation. Mention and discuss.

We have included the Calcium binding in cbb3 oxidases and modified Figure 3 to include a panel showing how calcium links heme b and heme b3 in cbb3 oxidase. We have also included two important references (Science, 329, 327-330 (2010) and Biochim Biophys Acta. 1817(6): 898–910 (2012)).

3. p.3 Why should a dimer be the active species? Is there any mechanistic reason? Please discuss!

We have added : Mutational study of AxqNOR (Glu494Ala) led to similar destabilisation of dimeric assembly and showed a monomeric cryoEM structure with mutated protein exhibiting a loss of enzymatic activity demonstrating that activity is disrupted by the loss of dimeric assembly of qNORs (8,9). The mechanistic advantage of a dimer is as yet unknown and as such we have refrained from further speculation.

4. p.4 There should be a ubiquinol binding site, but not a ubiquinone binding site. Binding of ubiquinone is detrimental because it would lead to substrate inhibition.

We have corrected this throughout.

5. p.5 Figure 1 legend: In EM you do not get "electron densities", electron densities are obtained by X-ray crystallography. Introduce the abbreviations DTM and LOP

We have corrected this and replaced electron density by "cryo-EM density map or cryo-EM map or cryo-EM map for the measured Coulomb potential of the scattered atoms". We have also spelled out DTM and LOP.

6. p.6 correct reductases, quinole, have to has, add "water" before W1

We have corrected these and carefully checked the manuscript any incorrect spelling.

7. p.7 correct comparision, Figure2f: replace bO3 by bo3, Cu is shown in orange not in blue, replace Cytochrome bo3 by cytochrome bo3, correct cytochtome

Done

8. p.8 correct Calcium, pseudomonas,

Done

9. p.9 correct coorectly, the electron donor is ubiquinol, not ubiquinone.

Done

10. The role of asp124 and his305 could be to take over the two protons from the ubiquinol which are released upon ubiquinol oxidation.

We thank the reviewer for raising this important point and have included a discussion on this at the end of the paragraph above Figure 4.

11. p.10 Figure 4C,D, I am confused: Why should electron transfer from the quinol to the heme b iron require a "pore" ???

We have corrected this.

12. p.11 introduce CuNiR, not mentioned before. What is its function, its relation to the qNOR? Correct chanel.

We have added a sentence with an additional reference introducing CuNiR and also CuNiR-NOR complex formation. Spelling check has been done.

13. p.13 figure legend correct chanel, Lauryl, Oleyl, indictive

Done

14. p.14 correct tranmembrane, remove of the structure, change has to have

Done

15. p.15 correct seperation, introduce DDM abbreviation,

Done

16. p.16 correct QNOR

Done

Other issues:

- C-terminus, and not C-terminal

This has been corrected

- No author contribution for Robert E. Eady

Done

- The correct term is electron cryomicroscopy, not cryo electron microscopy, because the electrons are not cryo, but the microscopy is done at cryo temperatures.

Response: We concur but electron cryo-microscopy cannot be abbreviated as cryo-EM (it should be literally abbreviated to 'E-cryoM'). We have changed the main text to "We have used single-particle electron microscopy with cryogenic sample stage (*cryo-EM*)" and put a footnote after cryo-EM (footnote : abbreviation has evolved to indicate that electron microscopy (EM) is done with sample maintained at cryogenic temperatures and distinguish it from the transmission electron microscopy without the cryogenic stage for the samples that are extensively used in material science research).

In our view neither terms 'cryo electron microscopy' or 'electron cryomicroscopy' are accurate as neither the electrons are cold or the microscope. It has evolved from insertion of frozen stage for pre-frozen samples (much like in protein crystallography where pre-frozen crystal samples are kept on a cryo-loop, but we do not call it frozen crystallography or frozen sample crystallography or cryo-crystallography).

We have consulted Dr Richard Henderson with the above difficulty who concurs that it needs resolving (Henderson and Hasnain are on the International Advisory board of the IUCrJ) and are going to approach the IUCr's Nomenclature Commission to resolve this for the long term benefit of the field.

Reviewer #2 (Remarks to the Author):

The manuscript by Flynn et al reports on a 2.2Å resolution structure of the qNOR from *Alcaligenes xylosoxidans*(Ax). The resolution is undoubtedly better than the (up to) 3.2 Å resolution structure of the same protein reported previously by the same group, but I do not find the quality of the analysis sufficient to warrant publication of the manuscript in its present form. I would like to add that the previous publications on the same or closely related proteins from the same group (their references 8 and 9) were clearly written, data well supported and conclusions reasonable and interesting, so it is unclear to me why the current manuscript lacks the same qualities.

My major general points are that it is unclear what exactly the higher resolution leads to in terms of new insights and that the analysis made is unclear and not supported by any complementary methods such as computational analysis, sequence analysis or mutational analysis. Many points made are unclear at best and at worst show misunderstanding about the heme-copper oxidase superfamily. The specific points made below are included to explain my concerns. It should also be noted that I cannot evaluate the technical quality of the cryo-EM analysis, which is hopefully covered by other reviewers.

Specific comments:

1. At the top of p.6 sequence conservation and evolutionary relationships to the heme-copper oxidases (HCOs) are described in a very unclear way, and no sequence alignments are included. What is conserved where? The Glu-490 mentioned is e.g. NOT conserved to the oxygen-reducing HCOs.

Even though Glu-490 is not conserved in the oxygen-reducing HCOs, but as shown in Figure 2F Glu-286 from a different part of the primary sequence is juxtaposed in a similar location.

2. p. 6. Last paragraph: The comparison to the E.coli bo3 O₂-reducing HCO is very unclear. Why focus on this here? The comparison when comes to quinol oxidation/binding is relevant but the bo3 active site is no more similar to qNORs than other members of the O₂ reducing HCOs are.

We have also included a comparison with the 3.2Å resolution structure of cbb3 oxidase also (Science 2010). We have added a figure S3 to show detailed comparison of AxqNOR, PacNOR and Ps cbb3 cox. The comparison with bo3 oxidase is justified on multiple grounds including, like qNOR (i) it has the auxiliary unit fused in the subunit containing the catalytic core, (ii) it lacks the redox centre in this fused subunit, (iii) receives its electron from quinol and is clearly a member of a family of quinol oxidases and (iv) its high resolution (~2.2Å) cryoEM structure was published recently.

Why would you expect a μ-oxo-bridge there? It has never been observed in any of the many structurally characterized O₂-reducing HCOs.

The high resolution of our structure clearly demonstrates the oxo-bridged metal site where the μ-oxo-bridge is primed for NO insertion. None of the other structures given in Figure 2 had a similar resolution or in the case of GsqNOR Fe was substituted by Zn generating an inactive enzyme. Metal substitution by Zn had placed it too distant from heme Fe to form a bridging link. A supplementary figure has been included with comparison of metal ion coordination of a typical binuclear metalloenzyme (hemerythrin and purple acid phosphatases, that were the first examples of μ-oxo, diiron protein to be crystallographically characterized). Other notable examples are methane monooxygenases and ribonucleotide reductase.

Also the sentence 'the conserved Glu (286 in this case)' is misleading since this is NOT the same Glu as in the NORs. This Glu (286 in bo3) is conserved to the A1-type HCOs only (sequence motif HPEVY), not to the B or C-type HCOs and NOT to the NORs (where the Glu mentioned is rather part of the different conserved HLWVE motif). The Glu mentioned in bo3 (286) sits at the end of a proton transfer pathway (D-pathway) to shuttle protons to the active site (and to be pumped) and fulfills a different role from the Glu (494 in Nm qNOR) in the NORs which is a ligand (at least during parts of the catalytic cycle) to the FeB. Why are these compared in this way? Sequence alignments, if supplied, would clearly show this.

See response to point 1. We have also provided detailed sequence comparisons in Figure S5.

And the same goes for the 'well-conserved Trp482' interacting with Glu-468. Here Glu-468 is not shown in the figure 2, but only much later (Fig 5). This should be clarified, sequence alignments provided (or at least clear reference to previous sequence alignments made) and Uniprot identifiers given as these sequences are not easy to find. Also, 'in each family member, Fe in heme b3..' is a bit misleading, as the heme is a heme o in the bo3.

We have modified Figure 2 to include Glu-468 (should have been Glu-478) and have changed heme o for bo3.

3. The Ca²⁺ discussion on p8 is unclear; why is this discussed here? What does the higher resolution provide that was not observed before in the same protein (ref 8)? The Ca site was identified in the previous structure already. Also, the C-type HCOs show similarity to the NORs in many respects such as the presence of a Ca ion in structurally the same position (see PDB ID 3MK7), but this is not mentioned in the list of other, much less related, Ca-containing metal proteins.

A paragraph has been added at the end of this section drawing parallel to Calcium binding site in the cbb3-type bacterial cytochrome c oxidases (cbb3-oxidases), the second most abundant cytochrome c oxidase group after the mitochondrial-like aa3-type cytochrome c oxidases.

Further on the same topic it is stated 'We postulate that electrons could be transferred through these residues (H629/H631)..'. Why? Electrons can tunnel, what is the basis for suggesting the transfer through specific residues at all in general and through these ones in particular? No support for this claim is presented. The continuation 'rather than solely through the Ca ion as suggested for cNOR' is even more unclear. I cannot find that this has ever been suggested for cNOR, the Ca is suggested to stabilize the hemes for efficient transfer directly between hemes (NOT through calcium which to my knowledge has never been observed to be redox-active). This section would need major revision.

We have revised this section in detail and have provided the necessary details (we are grateful to the reviewer for pointing out the inaccuracy due to the poor phrasing of the text, we apologise).

4. The quinol site discussion on p9-10 is in principle interesting. However the text does not correspond to the figure, modelling of the quinol is not clear (there is no picture with the model overlaid with the actual density so that the quality can be evaluated). And what would the native quinol donor be in Ax? Also the His305, Asp724 and Thr721 are said to 'form a neat pocket entrance', but apart from Asp-721 the fig 4B does not show how this would be accomplished, what is actually suggested to interact with the UQ molecule forming its binding site? (H305 I cannot see at all...).

Figure 4 has been modified to show all of the important residues to ensure that those mentioned in the text are clearly visible.

Also a RRY triad is suggested to interact with H₂O to make 'water-mediated electron transfer' feasible. Where are the references for such 'water-mediated eT'? Electrons generally prefer to tunnel through hydrophobic interiors where the reorganisation energy is low.

Electrons can transfer through bonds, tunnel through despite the energy barrier due to overlaps of potential diagrams etc but they can also be mediated via water as shown in several papers (e.g. Science 310, 1311–1313 (2005), Nature, 496, 123 (2013), NATURE COMMUNICATIONS | (2018) 9:5157 | DOI: 10.1038/s41467-018-07499-x). We have added a couple of these references.

Further, the discussion on the residues His305, Asp724 and Arg720 is also unclear; which ones are conserved also to the O₂-reducing quinol oxidising HCOs? And which ones to qNORs? Again, sequence alignments would clearly show this. If there are residues that occupy the same spatial location as in the O₂ reducing (quinol oxidising) HCOs but at a different (conserved) sequence location in qNORs, this would be interesting.

We have added the residue numbers for cytochrome bo₃, GsqNOR and our own structure.

5. The NO path discussion on p 10-11 is very unclear. On the top of p. 11 a possible binding to a NIR partner protein is mentioned without any background or reference to where and if this has been observed previously.

We have added a short paragraph and provided a reference.

The alternative route suggestion for NO is made without any support from calculations/tunnel analysis and is further suggested to contain water and charged and/or polar molecules, whereas all previous work in the area searched for hydrophobic pores/tunnels to accommodate gases like NO or O₂. These types of suggestions would need to be supported by analysis of sequence conservation and by feasibility in terms of computational and or experimental support.

The high resolution of the structure has allowed us to identify this alternative route which require testing among the family of HCO. These residues are well conserved in qNORs, as shown in the new Figure S5.

6. The product release path discussion on p 11-12 is also unclear. The first question is which product is discussed? Water or N₂O? There is then discussion of 'the' channel or pore without any introduction to what I presume is the water channel observed in previous qNOR structures suggested to be used for proton uptake. Here there is no mention/analysis of the possibility of proton uptake through this channel and no interpretation of what the differences to other structures would imply for the function. It is also not clear which details are better observed in this higher-resolution structure compared to the same authors' previous work.

We have provided additional details and contrasted with what was not observed in the previously reported wild type AxqNOR and Val494Ala mutant's structures due to limited resolution. The MD simulations of GsqNOR is referenced which had suggested 'channel enabling free exchange of water from the bulk water'.

7. The concluding remarks on p 14-15. Most of the text in this paragraph is rather a general introduction than a conclusion of the most important findings.

We have re-arranged text throughout the results so that key findings of the paper are highlighted under clearly defined headings. We have devoted the concluding remarks to make (a) general remarks in the first paragraph and (b) highlight the key findings that open up avenues for other researchers in the field to probe and test "these structurally informed putative routes for qNOR in combination with functional studies to establish factors that are important for molecular recognition, specificity and catalysis with wider implication to many redox enzyme systems including those of the respiratory heme-copper oxidase (HCO) superfamily".

Minor points

1. p5, top lists new features observed and refers to Fig 1B, where I can see no indication of 'putative nitric oxide entry channel'. Rephrase or show.

The figure has been modified.

2. p2, the sentence that starts : 'Three types of respiratory bacterial NOR are recognized' is unclear as it sounds as if the three types that are then mentioned belong to these three categories (which they don't); rephrase.

Done

3. p3, middle paragraph: 'These families have ahigh-spin heme b3'. Not correct, heme-copper oxidases can have heme a3, heme b3 or heme o3 in the active site.

Clarified.

4. p.6 middle paragraph refers to cNOR for Fig 2c, but it depicts GsqNOR.

Figure 2 has been revised.

5. Many places states 'Ca atom', 'Zn atom' etc when should be 'ion'.

We have now corrected this throughout the manuscript

6. There are many spelling mistakes, typos and incorrect use of italics which hinders the 'flow' of reading. A simple run through a spell-checker should help.

We have carefully checked the manuscript for all the spelling and character type.

Reviewer #3 (Remarks to the Author):

The article "A 2.2Å cryoEM structure of a quinol-dependent Nitric Oxide Reductase shows close similarity to the respiratory quinol oxidase family" by Alex J. Flynn et al. seeks to address how Quinol-dependent Nitric Oxide Reductases (qNORs) bind to quinol, and how they play a role in electron transfer. The 2.2Å cryo-EM structure of qNOR from *Alcaligenes xylosoxidans* provided informative details of the metalloproteins that are capable of defining key features that underpin catalysis involving redox sites.

In general, the article's results are solid, the conclusions are clear, and the discussions shed important light on respiratory heme-copper oxidase superfamily. I would be favorable for the publication of this manuscript. Some of their conclusions, statements or speculations may need citations of previous results to support their current analysis. Some figures may also need to be reorganized and recolored to highlight the contents. Given these potential shortcomings, it will be appreciated if this manuscript can be carefully revised before publication.

Major points:

Figure 1. B did not show the putative electron donor binding site and the nitric oxide entry channel. Although the water molecules could be seen, it's hard to define the critical waters.

The figure has been edited to show the key areas.

Figure 1. C, it is hard to find the Glu490. The author could either enhance the color or change the color to highlight this key residue—there are similar problems among all figures. "The μ -oxo bridge and newly identified water molecules linking to the conserved essential Glu490." This glutamic acid seems conserved among other family members. Providing a sequence alignment will be highly appreciated. It would be better if the authors can design point mutations to verify the water molecule linking function. Same for Trp482.

Figure 1 has been improved to address the above concerns, including clearly showing Glu490. Trp482 could not be included with density as it overcrowds the centre. Trp482 is already included in Figure 2 and Figure 5. Sequence alignments have been provided in Figure S5 (A, B, C)

Figure 3. C aligns the C alpha atoms by showing the similarity. Quantitative analysis for alpha distance is needed.

Figure 3 has been rearranged and now includes *Ps cbb3* cox site.

We have added the rms deviation of alpha distances in the text and caption to Figure S3.

Figure 4. A did not show Asp724 and His305, which have been mentioned in the main paragraph. Figure 4.D did not show the superimposition between NmQNO and AxQNO. Asp724 and His305 together with all the important residues mentioned in the text are shown. In panel A for AxQNO, we have modified it to include EM density for ubiquinol. Panel D now provides the superposition of Nm and Ax QNOs.

It is nice to obtain an alternative route for substrate access based on the high-resolution cryo-EM structure of AxQNO. If possible, I suggest the authors have biochemical or computational evidence to verify these results.

We agree with the reviewer that mutational analysis and computer simulations would further our understanding of this alternative pathway. However, this is an extensive amount of work both financially and in time and we feel that the results we have, are already impactful and do not wish to delay them. We are currently looking at obtaining funding to study this area but this is likely to be another high impact paper in itself.

“DTM replaced a bound lipid molecule during protein extraction”, but in the method part, the authors mentioned that the AxQNO was extracted with 1% DDM. Which one is right? Extracted the AxQNO with 1% DDM but purified with 0.05% DTM. Why choose DTM? In the 2021 paper, they used styrene–maleic acid co-polymer (SMA) nanodiscs and membrane scaffold protein (MSP) nanodiscs. Why did the authors choose different detergents? And does the DDM/DTM help to get the highest structure of NO?

In our 2019 Science Advances paper, we provided full details and is summarised in the purification section in the current manuscript is correct. DTM was chosen in final stages in an effort to obtain a higher resolution but it is possible that it may not be necessary. The Muench group have used both SMA and cyclic amphipoles for a range of membrane and we will investigate this for QNO. However, the grids used for this study were made pre 2021 and frozen down in liquid nitrogen. After seeing the significant improvements that can be gained from the new falcon 4 Selectris detector we collected a new data set as described here. We are yet to investigate if higher resolutions can be obtained with different extraction routes but given the high resolution we have, we do not feel that this is currently limiting the resolution of the protein and our activity assays show that DDM is not detrimental to activity.

Minor points:

Reference is needed to support “a weak density was observed in a similar position”.

We apologise for the oversight and have now added in the appropriate reference (Matsumoto et al., 2012)

Reference is needed to support “An important RRY triad was first discovered in cytochrome bo3...”

Sorry, this was incorrectly worded. This section has been modified.

Reference is needed to support “ Based on early crystallographic structures, Val 485 was considered important for guiding NO...” and “mutation of Val485Ala only reduced the activity only by 60%...”.

Reference added

Check whether it has a redundance “only”. “They can be clerly seen positioned within the transmembrane region of protein...” In this sentence, “clerly ” should be “clearly”.

Fixed as suggested.

REVIEWER COMMENTS

Reviewer #1 (Remarks to the Author):

Most of the criticism raised previously have been handled satisfactorily. Unfortunately the ms is still not perfect.

My major criticism is now that we have a different opinion on what a mu-oxo bridge is. In my opinion and that of many authors a mu-oxo bridge is formed by an O²⁻ and not a OH⁻. The figures and the manuscript have to be corrected!!! I also would consider that metal ion bridging waters are hydroxides and not waters.

The language style is still not perfect. There are also other points to be corrected:

line 135 apoferritin instead of apoferretin

general; write calcium and not Calcium

architecture and not architechure

legend of figure 4 (D) should be (C)

legend of figure 6 use caps in (a), (b), (c) as in figure

Reviewer #2 (Remarks to the Author):

Since my review contains some original points, as well as author responses and comments on them by me, it became hard to make that clear without proper formatting, so I have uploaded the file as pdf.

The revised manuscript by Flynn et al is improved compared to the initial submission. Some new issues have arisen and some of the initial issues remain as detailed below. It is also clear in the revised submission that the authors are focussed on the improved resolution per se, as indicated e.g. by the addition of the technical description on lines 120-141, and not as much on the insight it directly affords into the function of the qNORs. This is fine, it just raises the question about whether the manuscript is more suited to a specialized journal.

(My original comments in italics)

3. The Ca²⁺ discussion on p8 is unclear; why is this discussed here? What does the higher resolution provide that was not observed before in the same protein (ref 8)? The Ca site was identified in the previous structure already. Also, the C-type HCOs show similarity to the NORs in many respects such as the presence of a Ca ion in structurally the same position (see PDB ID 3MK7), but this is not mentioned in the list of other, much less related, Ca-containing metal proteins.

A paragraph has been added at the end of this section drawing parallel to Calcium binding site in the cbb3-type bacterial cytochrome c oxidases (cbb3-oxidases), the second most abundant cytochrome c oxidase group after the mitochondrial-like aa3-type cytochrome c oxidases.

Response: The addition of the comparison to the very similar Ca site in *cbb3* is good, but some issues remain: 1. There is still no clear description of what is actually new here? The Ca sites in NORs and their relation to the *cbb3*s has been described multiple times before. What new insight came from the higher resolution? (Minor point: Why call the carboxylates coordinated propionates in NORs but 'carboxylates of the pyrrole rings' in *cbb3* when they are the same?)

4. The quinol site discussion :

Further, the discussion on the residues His305, Asp724 and Arg720 is also unclear; which ones are conserved also to the O₂-reducing quinol oxidising HCOs? And which ones to qNORs? Again, sequence alignments would clearly show this. If there are residues that occupy the same spatial location as in the O₂ reducing (quinol oxidising) HCOs but at a different (conserved) sequence location in qNORs, this would be interesting.

We have added the residue numbers for cytochrome bo₃, GsqNOR and our own structure.

Response: On lines 319-320 there is still mention of an RRY triad for the Arg, Arg and Asn residues, why? One would think that the Y would be a Tyrosine.

The paragraph also talks of 'the' Arg720 in cytochrome bo₃, while it is not a conserved Arg (I think, it's not included in FigS5a, and there is no mention of S5b in the main text and it is unclear how to read it). It would be better to say that AxqNOR possesses 'a' critical Arg.

5. The NO path discussion on p 10-11 is very unclear. On the top of p. 11 a possible binding to a NIR partner protein is mentioned without any background or reference to where and if this has been observed previously.

We have added a short paragraph and provided a reference.

The alternative route suggestion for NO is made without any support from calculations/tunnel analysis and is further suggested to contain water and charged and/or polar molecules, whereas all previous work in the area searched for hydrophobic pores/tunnels to accommodate gases like NO or O₂. These types of suggestions would need to be supported by analysis of sequence conservation and by feasibility in terms of computational and or experimental support.

The high resolution of the structure has allowed us to identify this alternative route which require testing among the family of HCO. These residues are well conserved in qNORs, as shown in the new Figure S5.

Response: This section has improved, but is still unclear: 1) The references to possible complexes of NOR/NIR are still without clear references, presumably the co-complex of PacNOR-NIR (PNAS 2017) is where these things have been discussed in most details and

should be referenced. It is also said that the PacNOR was analysed without reference and without saying what was found to be similar. It says alternate paths were suggested here, but to my knowledge, they were also hydrophobic. Since there is still no experimental or computational support for this pathway, references to other systems where hydrophilic paths have been used for NO transfer are crucial. Showing that the residues are conserved (at least to the 3 qNOR sequences included) is a good first step, but it says nothing about what they are used for.

6. The product release path discussion on p 11-12 is also unclear. The first question is which product is discussed? Water or N₂O? There is then discussion of 'the' channel or pore without any introduction to what I presume is the water channel observed in previous qNOR structures suggested to be used for proton uptake. Here there is no mention/analysis of the possibility of proton uptake through this channel and no interpretation of what the differences to other structures would imply for the function. It is also not clear which details are better observed in this higher-resolution structure compared to the same authors' previous work.

We have provided additional details and contrasted with what was not observed in the previously reported wild type AxqNOR and Val494Ala mutant's structures due to limited resolution. The MD simulations of GsqNOR is referenced which had suggested 'channel enabling free exchange of water from the bulk water'.

Response: It is clear now that the authors mean for the product N₂O to be released through this water filled channel. Why would the gas N₂O rather be released through a water-filled channel than use the NO entry path? Support for using channels with these properties for N₂O release is missing. The authors also suggest the pathway could have an 'additional' role in proton transfer, but surely N₂O and H⁺ have very different properties and having them transfer through the same channel seems unlikely, do the authors have any support for this?

Minor points

6. There are many spelling mistakes, typos and incorrect use of italics which hinders the 'flow' of reading. A simple run through a spell-checker should help.

We have carefully checked the manuscript for all the spelling and character type.

There are still quite a few spelling/formatting mistakes, I list a few below:

Cbb₃ in FigS3, gene/protein names are CcoN O and P (not cox). Many instances hemes b3 without the subscript, FB instead of FeB, the word 'structure' missing on line 128, HQNQ instead of HQNO several times in Figure caption to Fig. 4

New minor point 1. On line 93, the Glu494Ala mutant is mentioned as if only affecting the monomer/dimer state, but to my knowledge, it also affects catalysis per se, clarify.

Reviewer #3 (Remarks to the Author):

The authors have addressed all my questions.

There is one thing I just felt a bit confused. It is the acknowledgement section. It looks to me that 'Samples were prepared by Chai Gopalasingam' should appear in the author contribution section rather than the acknowledgement. For most biological samples, it may take a considerable amount of time to prepare them. Some of the challenging targets may easily take years of efforts to simply optimize the most suitable conditions. I am not sure how straightforward the sample preparation was, or if there was any special consideration in this case, but I feel in most cases the persons who contributed to sample preparation should be listed as actual authors.

Response to reviewers;

Reviewer 1.

Most of the criticism raised previously have been handled satisfactorily. Unfortunately the ms is still not perfect.

1. My major criticism is now that we have a different opinion on what a mu-oxo bridge is. In my opinion and that of many authors a mu-oxo bridge is formed by an O²⁻ and not a OH⁻. The figures and the manuscript have to be corrected!!! I also would consider that metal ion bridging waters are hydroxides and not waters.

Response: As stated in International Union of Pure and Applied Chemistry (2005). Nomenclature of Inorganic Chemistry (IUPAC Recommendations 2005). Cambridge (UK): RSC-IUPAC. ISBN 0-85404-438-8. pp. 163–165 In naming a complex wherein a single atom bridges two metals, the bridging ligand is preceded by the Greek letter μ ,^[2] with a subscript number denoting the number of metals bound to the bridging ligand. μ_2 is often denoted simply as μ . The nature of the ligand could be oxide (OH⁻); hydroxide (O²⁻) but in all cases the metals are bridged by a single atom. Significant amount of spectroscopic data (Ligand-to-metal charge transfer LMCT) on NORs has been discussed in terms of spectroscopic properties being consistent with a dinuclear centre with a μ -oxo bridge between the two ferric ions and in which the proximal ligand is dissociated from the haem iron (see for example, J. Am. Chem. Soc. 120, 5147–5152 (1998), J. Biol. Chem. 279, 17120–17125 (2004) and Biochemical Society Transactions, 37- 392-399 (2009)).

The language style is still not perfect. There are also other points to be corrected:

line 135 apoferritin instead of apoferritin - Done

write calcium and not Calcium - Done

architecture and not architecture - Done

legend of figure 4 (D) should be (C) - Done

legend of figure 6 use caps in (a), (b), (c) as in figure - Done

Reviewer 2

1. The addition of the comparison to the very similar Ca site in cbb3 is good, but some issues remain: There is still no clear description of what is actually new here? The Ca sites in NORs and their relation to the cbb3s has been described multiple times before. What new insight came from the higher resolution? (Minor point: Why call the carboxylates coordinated propionates in NORs but 'carboxylates of the pyrrole rings' in cbb3 when they are the same?) The resolution afforded by the new structure means that we can now more accurately model the Ca binding site. It would be a remiss not to include this section in the manuscript. Additional details of Ca site in our 2.2 Å resolution structure has been included "High resolution AxqNOR structure shows holospheric coordination of Ca ion, different to previously determined structures (Table S2). Ca ion is ligated by 7 oxygens, OH Tyr78, O Gly76, OE2 and OE2 Glu407, single oxygens from propionates of heme b and heme b₃ and a water molecule; the bond length are shorter than previously determined and now ranged 2.1 to 2.5 Å. Water molecule was not visible in earlier determined structures due to limited resolution, Glu407 only contributed one bond, while propionate of heme b two bonds." A supplementary Table S2 with distance from Ca to the ligands is compared with previous low-resolution structures of AxqNOR, its mutant and cNOR clearly indicating more reasonable shorter metal-ligand distances.

2. On lines 319-320 there is still mention of an RRY triad for the Arg, Arg and Asn residues, why? One would think that the Y would be a Tyrosine. The paragraphh also talks of 'the' Arg720 in cytochrome bo3, while it is not a conserved Arg (I think, it's not included in FigS5a,

and there is no mention of S5b in the main text and it is unclear how to read it). It would be better to say that AxqNOR possesses 'a' critical Arg.

We have now corrected RRY to RRN and have rephrased this paragraph to reflect the reviewers' comments.

3. This section has improved, but is still unclear: 1) The references to possible complexes of NOR/NiR are still without clear references, presumably the co-complex of PacNOR-NiR (PNAS 2017) is where these things have been discussed in most details and should be referenced. It is also said that the PacNOR was analysed without reference and without saying what was found to be similar. It says alternate paths were suggested here, but to my knowledge, they were also hydrophobic. Since there is still no experimental or computational support for this pathway, references to other systems where hydrophilic paths have been used for NO transfer are crucial. Showing that the residues are conserved (at least to the 3 qNOR sequences included) is a good first step, but it says nothing about what they are used for.

We have added the reference and text "In denitrifying organisms, the fine-tuning of the levels of toxic NO has been suggested to involve product/substrate channelling from nitrite reductase (NiR) to NOR. Two genetically distinct types of nitrite reductase participate in denitrification, a cytochrome *cd₁* type, encoded by NirS and the more widely distributed copper-containing enzymes (CuNiR) encoded by NirK, as found in *A. xylosoxidans*. Both types of NiR have been shown to form a complex with c and q NORs and the crystal structure of a 2:1 (*cd₁*NiR)–(*PacNOR*)₂ complex from *P. aeruginosa* has been determined at 3.2Å resolution. The authors consider that due to the topology of the organization of these enzymes in the membrane, and consistent with all-atom molecular dynamics simulations, a 1:1 complex formed by coulombic interactions of NiR with the membrane, is more likely formed *in vivo*. The hydrophobic NO released from NiR preferentially diffuses into the membrane to bind to NOR accessing the active site through the hydrophobic NO transfer channel identified in the *PacNOR* structure. The retention of many structural features revealed from the crystal structures of qNORs and the catalytic domain of cNOR (see Fig S3A, B) suggest a similar scenario for NO channelling operates in *A. xylosoxidans*."

4. It is clear now that the authors mean for the product N₂O to be released through this water filled channel. Why would the gas N₂O rather be released through a water-filled channel than use the NO entry path? Support for using channels with these properties for N₂O release is missing. The authors also suggest the pathway could have an 'additional' role in proton transfer, but surely N₂O and H⁺ have very different properties and having them transfer through the same channel seems unlikely, do the authors have any support for this?

We have modified this section, changed the subheading to include both aspects i.e. potential product release channel and additional details of water channel. To substantiate further, we have conducted search for additional potential channels that may serve for product release from the active site, starting the search from binuclear site using CAVER and included a new figure S6.

5. There are still quite a few spelling/formatting mistakes, I list a few below: Cbb3 in FigS3, gene/protein names are CcoN O and P (not cox). Many instances hemes b3 without the subscript, FB instead of FeB, the word 'structure' missing on line 128, HQNQ instead of HQNO several times in Figure caption to Fig. 4

Done

6. New minor point 1. On line 93, the Glu494Ala mutant is mentioned as if only affecting the monomer/dimer state, but to my knowledge, it also affects catalysis per se, clarify.

The mutation causes the disruption of the dimeric assembly which we have shown is essential for catalytic activity (Science Advances 5, eaax1803 (2019) and IUCrJ 7, 404–415 (2020)). We have reworded this.

Reviewer 3

The authors have addressed all my questions.

There is one thing I just felt a bit confused. It is the acknowledgement section. It looks to me that 'Samples were prepared by Chai Gopalasingam' should appear in the author contribution section rather than the acknowledgement. For most biological samples, it may take a considerable amount of time to prepare them. Some of the challenging targets may easily take years of efforts to simply optimize the most suitable conditions. I am not sure how straightforward the sample preparation was, or if there was any special consideration in this case, but I feel in most cases the persons who contributed to sample preparation should be listed as actual authors.

The sample was prepared by Dr Gopalasingam as part of an earlier paper (Gopalasingam et al., Sci Adv. 2019), when multiple grids were made, some of which were frozen by Rachel Johnson (also an author of Sci. Adv. Paper). However, for this paper one of the grids was taken from the freezer and all data collection, data processing, analysis and manuscript preparation were conducted without Gopalasingam or Johnson, both graduated from our laboratories in 2019.

Dr Gopalsingam was offered to work on the new data in April 2022 and again when we saw him in Japan in June 2022 which he declined due to 'potential conflict of interest' resulting from his new position. During this time, he and Hasnain worked closely for a review in Current Opinion in Structural Biology (Hasnain had invited him to join as co-author) which appeared in July 2022.